# ICYM$^2$I: The illusion of multimodal informativeness under missingness

**Young Sang Choi**[1,*] **Vincent Jeanselme**[1,*] **Pierre Elias**[1,2]**, Shalmali Joshi**[1]

[1]Department of Biomedical Informatics, Columbia University, New York, NY, USA
[2]Seymour, Paul, and Gloria Milstein Division of Cardiology, Department of Medicine,
Columbia University Irving Medical Center, New York, NY, USA

## ABSTRACT

Multimodal learning is of continued interest in artificial intelligence-based applications, motivated by the potential information gain from combining different data modalities. However, modalities observed in the source environment may differ from the modalities observed in the target environment due to multiple factors, including cost, hardware failure, or the perceived *informativeness* of a given modality. This change in missingness patterns between the source and target environment has not been carefully studied. Naïve estimation of the information gain associated with including an additional modality without accounting for missingness may result in improper estimates of that modality's value in the target environment. We formalize the problem of missingness, demonstrate its ubiquity, and show that the subsequent distribution shift induces bias when the missingness process is not explicitly accounted for. To address this issue, we introduce ICYM$^2$I (In Case You Multimodal Missed It), a framework[1] for the evaluation of predictive performance and information gain under missingness through inverse probability weighting-based correction. We demonstrate the importance of the proposed adjustment to estimate information gain under missingness on synthetic, semi-synthetic, and real-world datasets.

## 1 INTRODUCTION

Multimodal learning is ubiquitous in machine learning as practitioners combine multiple data types to improve predictive performance in applications to healthcare (Perochon et al., 2023; Tu et al., 2024), robotics (Gao et al., 2024; Shah et al., 2023), and recommender systems (Chen et al., 2019). However, factors such as privacy concerns (Jaiswal & Provost, 2020; Zhang et al., 2021), cost-benefit tradeoffs of data-acquisition (Buck et al., 2010), and user preferences (Kossinets, 2006) imprint multimodal data with missingness. Additionally, even if modality complete data is available or curated at training, data noise (Cohen et al., 2004; Ma et al., 2023) and sensor failures (Inceoglu et al., 2021; 2023) may result in missing modalities in the target environment.

Although the missingness of modalities is a recurring challenge in real-world settings, current multimodal machine learning methods often assume that modalities are fully observed, both in source and target environments. When missingness is considered, the literature has focused on engineering efforts (Le et al., 2025; Wu et al., 2024) such as data selection (Hosseini et al., 2022), imputation (Tran et al., 2017; Cohen Kalafut et al., 2023; Malitesta et al., 2024), and architecture design (Chen et al., 2022; Zeng et al., 2022), which implicitly assume a stable missingness process between source and target environments. When this assumption is violated, the missingness mechanism induces a distribution shift (Zhang et al., 2023; Liu et al., 2023b) that biases the estimated informativeness of a given modality. Missingness is pervasive and impacts a broad range of application domains encountered in the multimodal literature: in breast cancer screening, biopsies are only performed if there are abnormal findings in a mammogram; in autonomous vehicles, LiDAR sensor dropout can occur due to weather and lighting conditions; and in online recommender systems, reviews are only

---

[*]Equal contribution. Corresponding author: young.sang.choi@columbia.edu.

[1]Code available on Github: https://github.com/reAIM-Lab/ICYM2I

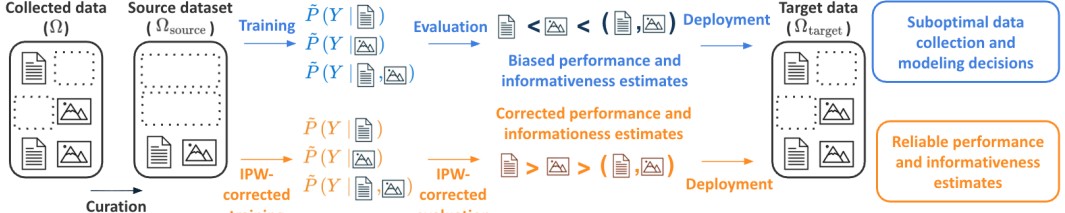

**Figure 1.** Overview of the proposed framework. Curation often discards missing data, resulting in a discrepancy between the collected $\Omega$ and source datasets $\Omega_{\text{source}}$ used for training. Current practice is denoted in blue: naïve training and evaluating on $\Omega_{\text{source}}$ leads to biased estimates of performance and informativeness on target data. The orange path illustrates the proposed ICYM$^2$I: a double inverse probability weighting (IPW) mechanism that yields accurate performance and informativeness estimates under the target distribution.

collected after certain consumer behaviors. Across these settings, ignoring the distribution shift between source and target due to missingness when quantifying modality informativeness may conflate missingness with signal, leading to flawed data collection and modeling decisions.

In this work, we propose a framework to overcome the (mis)estimation of both inherent informativeness and predictive utility under missingness in multimodal learning. Our contributions, summarized in Figure 1, are as follows:

• **Framework for multimodal learning with missingness.** We formalize the impact of missingness as a distribution shift *intrinsic* to multimodal learning, where the *observed* source distribution differs from the target distribution due to missingness. We show that not accounting for missingness, a common practice, may bias the estimate of a modality's predictive and information-theoretic utility.

• **ICYM$^2$I.** Under the missingness-at-random (MAR) assumption, a much more realistic assumption than the common and often implicit assumption of missingness-completely-at-random (MCAR) made by state-of-the-art multimodal strategies, we propose ICYM$^2$I (In Case You Multimodal Missed It), a double inverse-propensity weighting correction to overcome missingness-induced distribution shifts. The proposed method does not aim to improve predictive performance; rather, it enables unbiased estimation of model performance and information gain across modalities, even under missingness. Specifically, we demonstrate that ICYM$^2$I improves correlation in predictive and information-theoretic utility of modalities.

• **Experiments on diverse data.** We demonstrate the broad applicability and utility of our methods in synthetic, semi-synthetic, and real-world benchmark datasets, including a case study in multimodal learning in health.

## 2 RELATED WORK

**Multimodal benchmarks suppress missingness encountered in real-world environments.** Prior work on multimodal models often assumes *fully observed modalities* (Ngiam et al., 2011; Zadeh et al., 2017; Hou et al., 2019). Missingness has largely been an overlooked problem (Le et al., 2025; Wu et al., 2024), to the extent that current benchmarks rarely contain samples with missing modalities. Curation often involves discarding incomplete samples, filtering them based on data-quality criteria, such as text length or file size (Sharma et al., 2018; Schuhmann et al., 2022), or imputing them using automatic tools (Miech et al., 2019). This curation implicitly assumes that rejected samples follow the same distribution as the observed ones. This assumption may not hold. For example, in autonomous driving datasets, samples with sensor failures – often caused by extreme weather or lighting conditions – may be filtered out. Models trained on complete data may consequently not generalize to these settings, posing real-world risk at deployment. When missingness is considered, previous works focused on robustness through imputation (Tran et al., 2017; Cohen Kalafut et al., 2023; Malitesta et al., 2024), representation learning (Wu et al., 2024; Liu et al., 2023a), knowledge distillation (Li et al., 2024; Wang et al., 2020a), and model ensembling (Chen et al., 2022; Zeng et al., 2022) – all ignoring the potential shift resulting from the missingness process.

**Multimodal missingness in the target environment.** Prior work has explored missingness in the target environment (Lin & Hu, 2023; Zeng et al., 2022), e.g., when a captor fails at deployment (Ma et al., 2022). Broadly, two strategies have been proposed (Wu et al., 2024): (i) data preprocessing through cross-modal imputation (Cohen Kalafut et al., 2023; Malitesta et al., 2024; Tran et al., 2017), where one replaces the missing modality (Ma et al., 2021; Zhou et al., 2022), as well as (ii) model training strategies such as architecture design (Lee et al., 2023; Ge et al., 2023), distillation-based methods (Li et al., 2024; Wang et al., 2020a), and ensembling (Chen et al., 2022; Zeng et al., 2022). Through the proposed formalization, our work distinguishes between different missingness assumptions, demonstrating that the previously studied framework is only one among various plausible mechanisms for which current strategies are not well-designed.

**Distribution shifts in multimodal learning.** Addressing multimodal shifts has been studied in vision-language models (Zhou et al., 2024; Verma et al., 2024) or using information-theoretic notions to understand multimodal behavior under distribution shifts (Oh et al., 2025). Augmentation and regularization strategies have been leveraged to address temporal shifts for conversation understanding (Woo et al., 2023; Lian et al., 2023). Advances in learning, such as in-context learning, have been studied to characterize adaptation to multimodal distribution shifts (Zhou et al., 2024; Xue et al., 2024). However, existing strategies aim to improve robustness under domain shifts only, while ignoring the potential shift in missingness between source and target environments. Instead, *our work aims to correct estimates of performance and modality informativeness under missingness* to inform modality collection at deployment.

**Quantifying information-theoretic value of a modality.** Existing works often implicitly assume that additional modalities improve performance, ignoring the prohibitive cost, complexity, and potential noise added by these additional dimensions. When limited resources or constraints limit availability in the target environment, a central challenge is to quantify the information-theoretic value of a modality (Liang et al., 2024c). Liang et al. (2024a) proposed a method for recovering partial information decomposition measures of the redundancy, uniqueness, and synergy of the information provided by the different modalities (Bertschinger et al., 2014; Williams & Beer, 2010). However, these decompositions ignore the impact of missingness, a gap that our work aims to fill.

**Correcting for missingness bias.** The impact of missingness in multimodal learning is understudied, in part due to the lack of formalization of missingness in this area. Ignoring this process can bias estimates of interest (Phelan et al., 2017), because the observed distribution differs from the underlying distribution that practitioners aim to model. The statistical literature has introduced strategies such as matching (Stuart, 2010) and reweighting (Jethani et al., 2022) to address missingness. We leverage these statistical tools to address the overlooked missingness in multimodal settings.

## 3 MULTIMODALITY AND MISSINGNESS

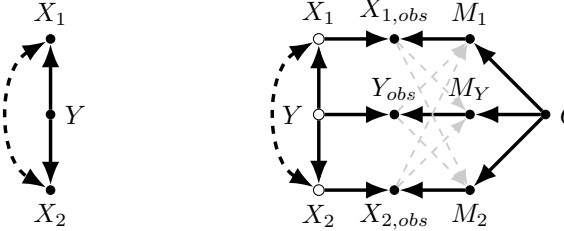

**Figure 2.** Directed Acyclic Graphs of the assumed data-generating processes. On the left is the commonly assumed graph with no missingness. On the right is the proposed missingness formalism. $X_1$ and $X_2$ are two modalities of interest, $Y$ is the label of interest. The missingness process depends on $C$. Filled point nodes are observed variables, while unfilled nodes are unobserved. Gray edges indicate MAR missingness for a given modality.

Consider two modalities, $X_1 \in \mathcal{X}_1$ and $X_2 \in \mathcal{X}_2$ and the state of interest $Y \in \mathcal{Y}$. We denote the joint underlying distribution $\Omega = \mathcal{X}_1 \times \mathcal{X}_2 \times \mathcal{Y}$. Without loss of generality, we assume an anti-causal setting for the data-generating process, in which the modalities are dependent on the states $Y$, as shown in Figure 2 (left). We use the binary indicators of missingness $M_1$, $M_2$, and $M_Y$, which are equal to 1 if the associated variable is missing, 0 if observed, following the convention of Mohan

& Pearl (2021). Observed variables are subscripted by 'obs', which corresponds to the underlying modality if observed, and unobserved otherwise (denoted by $\varnothing$). Formally, the observed variable $Y_{obs}$ and observed modalities $X_{1,obs}$ and $X_{2,obs}$ can be defined as follows:

$$Y_{obs} = \begin{cases} \varnothing & \text{if } M_Y = 1, \\ Y & \text{otherwise.} \end{cases}$$

In this setting, we denote the observed joint distribution $\Omega_{\text{obs}} := (M \cdot \mathcal{X}_1) \times (M \cdot \mathcal{X}_2) \times (M \cdot \mathcal{Y})$ where $M = M_1 \cdot M_2 \cdot M_Y$. This complete modalities analysis has been the focus of multimodal learning.

**Missingness in multimodal learning.** We distinguish three mechanisms that cover potential missing modality in multimodal settings (Rubin, 1976):

- Missing Completely At Random (MCAR): A modality is missing completely at random if the missingness process is independent of any other variable.
- Missing At Random (MAR): The missingness mechanism depends on observed variables only.
- Missing Not At Random (MNAR): Missingness depends on unobserved variables.

In Figure 2 (right), we describe the missingness mechanisms as dependent on $C$, a set of covariates that determine the missingness mechanism. Note that $C$ may include one of the modalities of interest, e.g., whether $X_2$ is observed may depend on the realization of $X_1$. In general, the set $C$ may differ for each modality depending on the data-generating process.

**Missingness-induced distribution shifts.** Missing modality $X_i$ and/or missing label $Y$ can induce distribution shifts between the source and the target distributions. For example, if a modality is observed in the target environment only if another one meets some criterion, then this distribution may not match the source distribution. Theoretically, we know that a non-MCAR missingness mechanism induces distribution shifts (Liu et al., 2023b; Zhang et al., 2023), i.e., the observed distribution differs from the underlying distribution. Critically, models trained and evaluated on the observed distribution are statistically biased estimates under any other missingness process.

For instance, consider an autonomous vehicle setting in which video and LiDAR are two modalities of interest. If the LiDAR sensor randomly fails, the missingness is MCAR. However, as previously mentioned, LiDAR may be more likely to malfunction under extreme weather conditions. If these conditions can be extracted from the video modality, one may assume MAR patterns. However, if the video cannot capture the variable that explains the LiDAR dysfunction (e.g., temperature), then LiDAR would be MNAR, as it is dependent on an unobserved variable. In the last two scenarios, focusing solely on complete samples would exclude all extreme-condition settings with critical real-world repercussions.

A common and often implicit assumption in the multimodal literature is the absence of missingness, which corresponds to either a MCAR mechanism ($\Omega_{\text{obs}} = \Omega$) or a stable missingness process between the source and target environments, i.e., the source and target distributions are the same ($\Omega_{\text{obs}}^{\text{source}} = \Omega_{\text{obs}}^{\text{target}}$). In other words, not adjusting for missingness assumes that the missingness process is uninformative or will remain the same in the target environment.

Prior works focus on improving the robustness of multimodal models when performance may degrade due to a modality missing in the target distribution, i.e, the source distribution reflects the true distribution while the target may present missingness $\Omega_{\text{obs}}^{\text{source}} \sim \Omega \neq \Omega_{\text{obs}}^{\text{target}}$.

Our work questions the applicability of these assumptions where modality collection is costly. While missingness may result in a distributional shift (Zhou et al., 2023), we emphasize that demonstrating the value of a modality in the source environment may lead to increased collection of this modality in the target environment, inducing a distribution shift akin to Assumption A.

**Assumption A** (Multimodal analysis informs data collection). *Demonstrated multimodal performance gain induces a shift in the missingness process in the target, i.e.* $\Omega_{obs}^{source} \neq \Omega^{target} \sim \Omega$.

We focus on settings in which historical data used to train a model are marked by missingness. Under such settings, we aim to: (i) identify which modalities are informative and may consequently be collected in the target environment, and (ii) train models using the observed source data that generalize to the target environment where the modalities are fully observed.

# 4 IS THIS MODALITY INFORMATIVE?

We aim to assess whether a partially missing modality would be informative if fully observed. To this end, we introduce $ICYM^2I$ (In Case You Multimodal Missed It), a framework for correcting model performance trained on modality complete samples where all modalities and labels are observed $(\Omega_{obs})$ to estimate the predictive utility of the partially missing modality if it were observed for the whole population $(\Omega)$. Additionally, we propose a correction to derive unbiased estimates of the information-theoretic utility of a modality, using $\Omega_{obs}$. We rely on Partial Information Decomposition (PID) (Williams & Beer, 2010) bounds introduced by Bertschinger et al. (2014) for this task, which quantifies the information value of a target of interest captured by two input variables.

**Correction.** We propose an Inverse Probability Weighting (IPW) approach (Robins et al., 1994), which reweights samples according to their probability of being observed. Under Assumption B that relaxes the common MCAR assumption made in the multimodal literature, IPW recovers unbiased estimates of the true distribution, enabling learning and evaluation on the true distribution from observed samples. IPW-adjustment is critical for both training and evaluation of multimodal models under missingness. IPW-adjusted training results in a model trained to infer on the underlying distribution $\Omega$, while correction of the evaluation allows for measuring performance on $\Omega$, despite evaluating the model only on samples from the observed distribution $\Omega_{obs}$.

**Assumption B** (MAR and Positivity). *The missingness mechanism is MAR, and $p_\Omega(M_1 = 0, M_2 = 0, M_Y = 0|C) > 0$.*

## 4.1 A MOTIVATING EXAMPLE

We consider the common multimodal example of learning bit-wise logic operators (Bertschinger et al., 2014; Harder et al., 2013; Liang et al., 2024a). We generate $10,000$ points with two modalities drawn from Bernoulli distributions ($p = 0.5$). The output state $Y$ is defined using the binary operators AND, OR, and XOR of input bits $X_1$ and $X_2$. In this setting, we induce missingness $M_2$ in $X_2$ and $Y$ as a function of $X_1$ (MAR): $M_2 \sim Bern(0.6X_1 + 0.2)$, resulting in 50% missingness in $X_2$. We investigate the impact of missingness on current strategies for evaluating the predictive and information-theoretic utility of a given modality.

**Estimating performance for informativeness.** A common practice to measure the predictive value of adding a modality is through ablation studies where practitioners train models using different modalities on the subset of samples where all modalities are observed $(\Omega_{obs})$. First, unimodal models $f(x_i)$ are trained to approximate $p_{\Omega_{obs}}(y \mid x_i)$, $\forall i \in \{1, 2\}$ and a multimodal model $f(x_1, x_2)$ to approximate $p_{\Omega_{obs}}(y \mid x_1, x_2)$ on the same observed dataset. The performance is then compared in a hold-out set, sampled from $\Omega_{obs}$. The informativeness of a modality is measured by the relative performance gain of the multimodal model over the unimodal model. However, multimodal models can underperform their unimodal counterparts due to data characteristics (Zhang et al., 2024) and learning dynamics (Wang et al., 2020b; Zhai et al., 2024). Thus, relying solely on performance as a proxy for informativeness, particularly under distribution shifts, can be misleading.

**Partial Information Decomposition (Bertschinger et al., 2014).** As an alternative to estimating performance, existing works have decomposed the informativeness associated with each modality (Liang et al., 2024a). Bertschinger et al. (2014) formalized this decomposition by analyzing the total (three-way) mutual information $I(Y : (X_1, X_2))$ (McGill, 1954; Te Sun, 1980), a measure of dependency between the target variable $Y$ and the modalities $(X_1, X_2)$. This quantity can be decomposes into four components: shared information (information both $X_1$, $X_2$ share about $Y$), unique information 1 (information only $X_1$ has about $Y$), unique information 2 (information only $X_2$ has about $Y$), and complementary information (information about $Y$ that requires both $X_1$ and $X_2$) as follows:

$$I(Y : (X_1, X_2)) = \underbrace{SI(Y : X_1; X_2)}_{\text{shared information}} + \underbrace{UI(Y : X_1 \backslash X_2)}_{\text{unique information 1}} + \underbrace{UI(Y : X_2 \backslash X_1)}_{\text{unique information 2}} + \underbrace{CI(Y : X_1; X_2)}_{\text{complementary information}}$$

Bertschinger et al. (2014) specifies how to estimate these quantities (see Appendix B); for instance, the unique information between $Y$ and $X_1$ can be estimated using the following:

$$\widetilde{UI}(Y : X_1 \backslash X_2) = \min_{q \in \Delta_\Omega} [I_q(Y : (X_1, X_2)) - I_q(Y : X_2)],$$

where $\Delta_\Omega$ is the set $q(X_i = x_i, Y = y) = p_\Omega(X_i = x_i, Y = y) \; \forall x_i \in \mathcal{X}_i, y \in \mathcal{Y}, i \in \{1, 2\}$, that is, the set of joint distributions over $(X_1, X_2, Y)$ that match the true two-way data distributions. Note that the objective function requires minimization with respect to the three-way mutual information. Importantly, Bertschinger et al. (2014) demonstrates that the solution to any one objective specifies an optimum solution for all other quantities.

Prior work that relies on this Partial Information Decomposition (PID) to attribute information-theoretic value implicitly assumes that $\Omega_{\text{obs}}^{\text{source}} = \Omega^{\text{source}} = \Omega^{\text{target}} = \Omega$. Instead, we evidence the limitations of these strategies performed on $\Omega_{\text{obs}}^{\text{source}} \neq \Omega$ when the target decomposition is $\Omega^{\text{target}} = \Omega$, i.e., the true data-generating mechanism.

**Table 1.** Impact of missingness on multimodality information for bitwise logic operators. Parentheses denote standard deviation across batches.

| | | AUROC | | | Information Decomposition | | | |
|---|---|---|---|---|---|---|---|---|
| | | $X_1$ | $X_2$ | $X_1 + X_2$ | Unique 1 | Unique 2 | Shared | Complementary |
| AND | Oracle | 0.83 (0.01) | 0.84 (0.01) | 1.00 (0.00) | 0.05 (0.00) | 0.03 (0.00) | 0.26 (0.00) | 0.47 (0.00) |
| | Observed | 0.66 (0.01) | 0.93 (0.01) | 1.00 (0.00) | 0.44 (0.00) | 0.00 (0.00) | 0.15 (0.00) | 0.36 (0.00) |
| | ICYM²I | 0.83 (0.01) | 0.85 (0.02) | 1.00 (0.00) | 0.03 (0.00) | 0.03 (0.00) | 0.27 (0.00) | 0.45 (0.00) |
| OR | Oracle | 0.84 (0.01) | 0.83 (0.01) | 1.00 (0.00) | 0.04 (0.00) | 0.05 (0.00) | 0.27 (0.00) | 0.46 (0.00) |
| | Observed | 0.95 (0.01) | 0.77 (0.01) | 1.00 (0.00) | 0.01 (0.00) | 0.15 (0.00) | 0.10 (0.00) | 0.23 (0.00) |
| | ICYM²I | 0.85 (0.02) | 0.82 (0.01) | 1.00 (0.00) | 0.03 (0.00) | 0.02 (0.00) | 0.27 (0.00) | 0.50 (0.00) |
| XOR | Oracle | 0.51 (0.02) | 0.49 (0.01) | 1.00 (0.00) | 0.00 (0.00) | 0.00 (0.00) | 0.00 (0.00) | 0.99 (0.00) |
| | Observed | 0.52 (0.02) | 0.80 (0.02) | 1.00 (0.00) | 0.34 (0.00) | 0.07 (0.00) | -0.07 (0.00) | 0.62 (0.00) |
| | ICYM²I | 0.53 (0.03) | 0.49 (0.03) | 1.00 (0.00) | 0.00 (0.00) | 0.00 (0.00) | 0.01 (0.00) | 0.96 (0.00) |

As a motivating example, we analyze the impact of missingness on estimating PID for unidimensional modalities with a bitwise-logic outcome (AND, OR, and XOR). Table 1 (left) presents the discriminative performance associated with neural networks trained on each individual modality and their combination under three scenarios: (i) access to all data (**Oracle**), (ii) focusing only on datapoints with all covariates observed (**Observed**), and (iii) adequately accounting for missingness (ICYM²I using IPW to adjust $\Omega_{\text{obs}} \mapsto \Omega$, by modeling the missingness mechanism), as proposed in Section 4.2. Table 1 (right) presents PID, discussed in Section 4.3 under the same scenarios, demonstrating how information decomposition is also biased due to missingness.

Specifically, relying on $\Omega_{\text{obs}}$ overestimates the performance of $X_1$ for OR but underestimates it for AND. Similarly, biased decomposition overestimates the informativeness of $X_1$ ("Unique 1" compared to "Unique 2") for OR. As $X_1$ informs the missingness process, it indirectly informs the outcome of interest, despite the true underlying generative process being dependent on both. The use of IPW can correct for such bias under positivity, provided the IPW propensities can be estimated (i.e., the MAR assumption). We study sensitivity to this assumption in Appendix D, where we further evaluate the robustness of our method under MCAR and MNAR, demonstrating robustness under MCAR.

We now formally describe two methods for reliably inferring the informativeness of modalities using (i) unbiased estimation of unimodal versus multimodal model performance using supervised learning (ICYM²I-learn), and (ii) high-dimensional autodifferentiable partial information decomposition (ICYM²I-PID). In addition, we demonstrate the need for IPW-adjusted *evaluation* as a key element to determine modality informativeness using supervised learning.

## 4.2 ICYM²I-LEARN: ESTIMATING PREDICTIVE PERFORMANCE UNDER MISSINGNESS

**Training.** Under the MAR assumption, i.e., the missingness is fully explained by observed covariates $C$; that is, the probability of a data point being missing depends only on $C$, we propose to train the model with a weighted loss using samples from $\Omega_{\text{obs}}$. The proposed IPW-adjusted loss accounts for the distributional shift ($\Omega_{\text{obs}} \mapsto \Omega$) by up-weighting under-observed samples, as described in the following lemma.

**Lemma 1** (IPW Training). *The loss function computed on the observed data $l_{\Omega_{obs}}(x_1, x_2, y)$ can be reweighted to approximate the target loss $l_\Omega(x_1, x_2, y)$ as follows:*

$$l_\Omega(x_1, x_2, y) = \frac{1}{1 - p(m_1, m_2, m_y \mid C)} \, l_{\Omega_{obs}}(x_1, x_2, y)$$

*where $p(m_1, m_2, m_y \mid C)$ is the probability of missingness, given the covariates $C$.*

**Evaluation.** Existing work suffers from an analogous bias in model evaluation, by relying on a hold-out set from the observed distribution ($\Omega_{\text{obs}}$). To estimate a given metric under the true underlying distribution, one must correct this metric using a similar correction as previously described. For instance, Li et al. describes how to correct for both AUC and Brier score using IPW.

**Corollary 1** (ICYM$^2$I-learn). *Consider a model $f$ trained and evaluated on data drawn from $\Omega_{obs}$. To correct the model and estimate its performance on $\Omega$, one must correct* both *its training and evaluation following the previous corrections.*

## 4.3 ICYM$^2$I-PID: PARTIAL INFORMATION DECOMPOSITION FOR MULTIMODAL INFORMATIVENESS

Obtaining the partial information decomposition bounds corresponds to estimating the distribution $q \in \Delta_\Omega$ that minimizes three-way mutual information (see Bertschinger et al. (2014), Lemma 4), where $\Delta_\Omega$ is the set of all distributions over $(X_1, X_2, Y)$ such that the two-way joints between $X_i$ and $Y$ match the true data-generating distribution. Formally, one can focus on solving:

$$\min_{q \in \Delta_\Omega} I_q(Y : (X_1, X_2))$$

To operationalize this minimization, Liang et al. (2024a) proposes to minimize this quantity using the observed samples:

$$\Delta_\Omega \approx \{q \propto \exp(f_1(x_1) \cdot f_2(x_2)) : q(x_i, y) = p_{\Omega_{\text{obs}}\phi}(x_i, y) \, \forall x_i \in \mathcal{X}_i, y \in \mathcal{Y}, i \in \{1, 2\}\}$$

where $p_{\Omega_{\text{obs}}\phi}$ is a re-parametrization of $\Omega_{\text{obs}}$ using neural networks. To enforce the marginals of $q$ to match $p_{\Omega_{\text{obs}}}$, Liang et al. (2024a) proposes an iterative process using a Sinkhorn–Knopp procedure (Knight, 2008).

Under missingness, we observe samples from $\Omega_{\text{obs}}$ rather than $\Omega$. To unbias the PID measures in this setting requires adjusting for the $\Omega_{\text{obs}} \mapsto \Omega$ shift. Effectively, our approach corrects the previous optimization to account for the distribution shift induced by MAR missingness.

First, we introduce a corrected mutual information computation to ensure that we optimize an unbiased estimate of the three-way mutual information under $\Omega$ using samples from $\Omega_{\text{obs}}$ (see Appendix A for the complete proof).

**Lemma 2** (Corrected mutual information).

$$I_\Omega^{\text{IPW}}(Y : (X_1, X_2)) = \mathbb{E}_{\substack{x_1, x_2 \sim p_{\Omega_{obs}}(x_1, x_2) \\ y \sim p_\Omega(y|x_1, x_2)}} \left[ \frac{1 - p(m_1, m_2)}{1 - p(m_1, m_2|x_1, x_2, y)} \log \left( \frac{p_\Omega(x_1, x_2, y)}{p_\Omega(x_1, x_2) p_\Omega(y)} \right) \right]$$

Then, ICYM$^2$I-PID uses the following projection set to operationalize PID-bound estimation while accounting for missingness:

$$\Delta_\Omega^{\text{ICYM}^2\text{I}} \approx \{q \propto \exp(f_1(x_1) \cdot f_2(x_2)) : q(x_i, y) = p_{\Omega\phi}(y, x_i) \, \forall x_i \in \mathcal{X}_i, y \in \mathcal{Y}, i \in \{1, 2\}\}$$

$$= \{q \propto \exp(f_1(x_1) \cdot f_2(x_2)) : q(x_i, y) = \text{IPW}_{p_{\Omega\phi}}(p_{\Omega_{\text{obs}}\phi}(y, x_i)) \, \forall x_i \in \mathcal{X}_i, y \in \mathcal{Y}, i \in \{1, 2\}\}$$

where $\text{IPW}_q(p)$ is an IPW correction for the shift $p \mapsto q$ using samples from $p$ and $p_\phi$ is a re-parametrization of $\Omega$ using neural networks and learned using samples from $\Omega_{\text{obs}}$ via the weighted loss introduced in Lemma 1. Importantly, the proposed correction is agnostic to the parametrisation of $q$. We note that these mutual information-based quantities can be equivalently estimated using entropy-based measures (see derivations in Appendix B).

Estimating PID under missingness therefore consists in optimizing:

$$\min_{q \in \Delta_\Omega^{\text{ICYM}^2\text{I}}} I_q^{\text{IPW}}(Y : (X_1, X_2))$$

Similarly to Liang et al. (2024a), the unbiased PID estimation framework consists of the following steps (detailed in Appendix C):

1. **Model the missingness mechanism**. Train a model to estimate the probability of missingness given $C$ to obtain IPW weights for correcting the distribution shift.

2. **Train corrected unimodal and multimodal models**. Train each model ($f_1(x_1)$, $f_2(x_2)$, and $f(x_1, x_2)$) using the IPW-corrected loss introduced in Lemma 1. These models can be trained with flexible inductive biases tailored to the application at hand.

3. **Solve PID optimization**. Estimate $q \in \Delta_\Omega^{\mathrm{ICYM^2I}}$ minimizing $I_q(Y : (X_1, X_2))$, where $q$ is parameterized by the product of two unimodal networks. To match the marginal of $q$ to the estimated probabilities of the corrected unimodal neural networks, we apply a modified Sinkhorn–Knopp procedure (Knight, 2008) using IPW-corrected unimodal distributions[2]. Our framework is agnostic to the choice of parametrization of $q$, provided that calibrated probabilistic scores are learned.

4. **Estimate the PID components**. Given $q$, compute PID quantities using the bounds of Bertschinger et al. (2014), corrected via the IPW-correction introduced in Appendix A.

## 5 EXPERIMENTS

To better understand the connection between performance, information decomposition, and missingness, we propose a simulation (detailed in Appendix E), two semi-synthetic studies that reflect real-world missingness mechanisms (see Appendix F), and a real-world case study.

### 5.1 SIMULATION AND SEMI-SYNTHETIC EXPERIMENTS

**Table 2.** Comparison between estimated AUC performance under the different training and evaluation corrections and oracle performance on $\Omega$. $\epsilon$ denotes the RMSE between estimated and oracle AUCs.

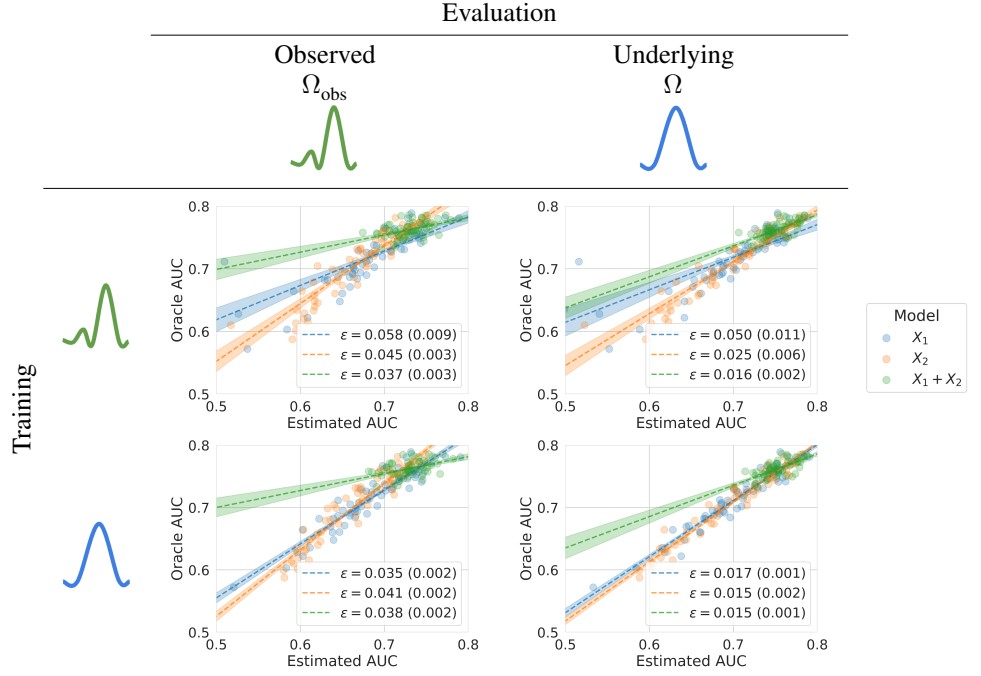

In Table 2, each point represents the estimated performance value for one simulation under the training and evaluation IPW-corrections and the oracle performance, i.e., a model trained and tested

---

[2]Prior work typically matches $q(y \mid x_i)$ to $\Omega_{\mathrm{obs}}(y \mid x_i)$, which yields biased PID estimates under missingness.

on $\Omega$. Specifically, columns reflect evaluation correction, while rows reflect training correction. These results underline the importance of correcting both training and evaluation, as proposed in $ICYM^2I$, to best align with the performance one would obtain on $\Omega$, as shown by the smallest Root Mean Squared Error (RMSE) observed when both corrections are applied. This observation shows that the proposed $ICYM^2I$ best estimates the performance associated with each modality, despite relying on $\Omega_{obs}$. Appendix E echoes the same observation when estimating PID.

The semi-synthetic experiments examine the effect of enforcing increasing missingness on the performance and information decomposition of UR-FUNNY (Hasan et al., 2019) and hateful memes (Kiela et al., 2020), two foundational real-world datasets used in the multimodal literature for affective computing and content moderation. Table 3 summarizes the effect of enforcing 70% missingness on estimating multimodality informativeness across these datasets, demonstrating the generalizability of our proposed strategy across real-world datasets. Appendix F further illustrates the robustness of the methodology under different levels of missingness in these datasets and explores MNAR patterns.

**Table 3.** Impact of 70% missingness on multimodality information for UR-FUNNY (Hasan et al., 2019) and Hateful Memes (Kiela et al., 2020). Parentheses denote standard deviation across batches.

| | | AUROC | | | Information Decomposition | | | |
|---|---|---|---|---|---|---|---|---|
| | | Text | Image/Video | Image + Text | Unique$_{text}$ | Unique$_{image}$ | Shared | Complementary |
| UR-FU. | Oracle | 0.68 (0.01) | 0.60 (0.02) | 0.69 (0.02) | 0.10 (0.00) | 0.02 (0.00) | 0.00 (0.00) | 0.00 (0.00) |
| | Observed | 0.61 (0.03) | 0.54 (0.04) | 0.63 (0.03) | 0.05 (0.00) | 0.00 (0.00) | 0.03 (0.00) | 0.00 (0.00) |
| | $ICYM^2I$ | 0.66 (0.03) | 0.57 (0.04) | 0.62 (0.04) | 0.07 (0.00) | 0.00 (0.00) | 0.00 (0.00) | 0.00 (0.00) |
| Memes | Oracle | 0.71 (0.01) | 0.57 (0.01) | 0.72 (0.01) | 0.09 (0.01) | 0.00 (0.00) | 0.04 (0.00) | 0.05 (0.01) |
| | Observed | 0.68 (0.02) | 0.61 (0.02) | 0.71 (0.02) | 0.13 (0.00) | 0.04 (0.00) | 0.01 (0.00) | 0.00 (0.00) |
| | $ICYM^2I$ | 0.67 (0.02) | 0.61 (0.02) | 0.71 (0.02) | 0.10 (0.00) | 0.01 (0.00) | 0.02 (0.03) | 0.03 (0.01) |

## 5.2 CHEST RADIOGRAPHS ARE UNINFORMATIVE OVER ELECTROCARDIOGRAMS FOR STRUCTURAL HEART DISEASE DETECTION.

While our core contribution is methodological, this section illustrates how ignoring missingness can lead to biased estimates of the informativeness of a given modality in a real-world setting where modalities are commonly missing. Specifically, we study structural heart disease (SHD), a set of conditions that affect the heart's physiology, which is typically diagnosed using transthoracic echocardiograms (TTEs) (Writing Committee Members et al., 2021). However, TTEs are often underutilized in the United States due to diagnostic stewardship and competing financial incentives (Papolos et al., 2016). Prior work using unimodal models with common modalities in electrocardiograms (ECGs) (Elias et al., 2022; Ulloa-Cerna et al., 2022) and chest radiographs (CXRs) (Bhave et al., 2024) has demonstrated that non-TTE modalities can detect structural heart disease labels. However, CXRs are not systematically collected in conjunction with ECGs, leading to systematic missingness patterns. We therefore evaluate $ICYM^2I$ on this clinical task to assess the informativeness of CXRs for diagnosing SHD, despite their missingness.

**Dataset.** Our study population comprises a retrospective cohort of 98,397 adult patients who received an ECG and a TTE within 1 year of each other. The population has 20.56% SHD prevalence. In this cohort, 12,587 patients (12.79%) have recorded CXRs. For subjects with multiple echocardiograms, we select the first TTE to model opportunistic screening with non-TTE modalities. All data were collected from Columbia University Medical Center, a large academic medical system in New York City, between 2008 and 2022. Data are split temporally, with subjects with TTEs collected on or after 2018 ($n = 40,734$) allocated to the test set. All data were de-identified, retrospective, and collected for clinical purposes within an academic hospital system, with Institutional Review Board approval. Appendix G contains further details regarding preprocessing, embedding generation, and the $ICYM^2I$ implementation.

**Results.** Table 4 presents the performance of each uni- and multimodal model, along with the associated information decomposition. While both the observed and corrected analyses demonstrate the importance of ECG in modeling SHD, the corrected results raise questions about the information gain associated with CXR. Naive decomposition suggests the unique information in CXRs accounts

for about 5% of the total information. However, ICYM$^2$I reduces this unique contribution to 1.8% while increasing estimates of shared information between ECG and CXRs for SHD detection. In contrast to domain knowledge, where ECGs capture electrophysiology while CXRs capture structure and anatomy, two distinct aspects of cardiac health, the corrected complementary and shared results, and low unique information of CXRs suggest that CXRs are not independently useful for SHD diagnosis. Note that our results indicate that the multimodal model performs slightly worse than the unimodal ECG model, reflecting the potential risk of overfitting associated with a large number of features.

**Table 4.** Informativeness of ECG and CXR modalities on model-based structural heart disease detection. Parentheses denote standard deviation across batches ($n = 1024$).

| | AUROC | | | Information Decomposition | | | |
|---|---|---|---|---|---|---|---|
| | ECG | CXR | ECG + CXR | Unique$_{ECG}$ | Unique$_{CXR}$ | Shared | Complementary |
| Observed | 0.83 (0.01) | 0.72 (0.02) | 0.82 (0.01) | 0.11 (0.00) | 0.01 (0.00) | 0.10 (0.00) | 0.00 (0.00) |
| ICYM$^2$I | 0.82 (0.01) | 0.73 (0.02) | 0.83 (0.01) | 0.07 (0.00) | 0.01 (0.00) | 0.48 (0.00) | 0.01 (0.00) |

## 6 DISCUSSION

This work formalizes the issue of missing modalities in multimodal settings. We emphasize that existing work commonly overlooks missingness by discarding samples with any missing modality during curation, or implicitly assumes that the missingness mechanism remains stable when a model is deployed in the target environment. Our work formalizes this problem and demonstrates its ubiquity in the multimodality literature. Most critically, prior work ignores that any perceived informativeness of a modality may result in increased rates of data collection, inducing different missingness patterns at deployment. Our work, therefore, introduces ICYM$^2$I, a correction to estimate the information gain associated with a *partially observed modality*. Our results demonstrate the methodology's capacity to correct for biases introduced by missingness across synthetic, semi-synthetic, and real-world multimodal datasets. Finally, we demonstrate the practical utility of this methodology on a healthcare dataset, showing the divergent conclusions that one would reach if missingness were ignored. Our work highlights the critical importance of missingness in multi-modal research and urges practitioners to pay particular attention to this issue by systematically *collecting* data with incomplete modalities and carefully *modeling* and *accounting* for missingness to enhance robustness.

**Limitations.** The key assumption in our work is that missingness is MAR. No theoretical guarantees exist under MNAR patterns. While distinguishing these assumptions is empirically untestable, practitioners should ensure that this assumption is appropriate for their data. Importantly, MAR is less restrictive than the implicit MCAR assumption made in the multimodal literature, and does not require distributional assumptions that one must assume to tackle MNAR patterns. Additionally, our work is based on Partial Information Decomposition (PID) (Bertschinger et al., 2014), which focuses on two input modalities. In practice, practitioners could consider a one-vs-all approach to inform modality informativeness using our method. However, extending the decomposition to more than two modalities remains an open challenge (Griffith & Koch, 2014; Kolchinsky, 2022), where notions of mutual information are not well-defined beyond three-way mutual information. As in prior work on PID-based measures of information gain on high-dimensional data (Liang et al., 2023; 2024a), the quality of the representations used may impact the measures returned by ICYM$^2$I. We ensure that our probabilistic estimates are calibrated to mitigate such challenges. Additionally, PID is designed for datasets with paired modalities, where the input modalities are associated with an instance (e.g., a patient and their associated chest X-ray and electrocardiogram); consequently, our method does not extend to unpaired settings. Finally, our proposed method relies on a notion of instance for which all modalities are observable. Extending this method when there is no notion of an instance, i.e., unaligned modalities, could be considered but would require different inductive biases to model the underlying unimodal and multimodal probabilities.

**Ethics statement.** Our work demonstrates the impact of missingness on performance estimates in multimodal learning. We demonstrate the utility of our method in a crucial healthcare use case. However, the methodology remains a proof of concept that would require additional testing to be deployed in a real-world context. Our study is approved by the Columbia University Medical Center Institutional Review Board (IRB-AAAU7973). We have extracted all data in HIPAA-compliant servers, and our experiments are also conducted on HIPAA-compliant compute despite being deidentified for extra caution. While beyond the scope of this work, modality completeness is not uniform across demographic subgroups and can manifest in data collection policies, such as differential access to care based on insurance status. Our method could provide important insights into the utility of multimodal predictions in such settings.

**Reproducibility statement.** Theoretical proofs are provided in Appendix A. All code for applying the proposed ICYM²I and reproducing all synthetic and semi-synthetic results presented in this work is publicly available on Github[3]. A summary of the computational resources required to reproduce our results is given in Appendix H.

## ACKNOWLEDGMENTS

VJ and SJ acknowledge partial support from NIH R01MH137679. YSC and SJ acknowledge partial support from Google Research Scholar Awards. SJ acknowledges partial support from the SNF Center for Precision Psychiatry and Mental Health at Columbia. Any opinions, findings, conclusions, or recommendations in this manuscript are those of the authors and do not reflect the views, policies, endorsements, expressed or implied, of any aforementioned funding agencies/institutions.

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

## A  PROOFS

This section provides the proofs for Lemma 1 and Lemma 2.

**Lemma 1.** *The separable loss function computed on the observed data $l_{\Omega_{obs}}(x_1, x_2, y)$ can be reweighted to approximate the target loss $l_\Omega(x_1, x_2, y)$ as follows:*

$$l_\Omega(x_1, x_2, y) = \frac{1}{1 - p_\Omega(m_1, m_2, m_y \mid C)} \, l_{\Omega_{obs}}(x_1, x_2, y)$$

*where $p_\Omega(m_1, m_2, m_y \mid C)$ is the probability of missingness, given the covariates $C$.*

*Proof.* The proof is analogous to that of Lemma 2, which we show in detail, for any separable loss function $l(x_1, x_2, y)$. $\qquad\square$

**Lemma 2.**

$$I_\Omega(Y : (X_1, X_2)) = \mathbb{E}_{\substack{x_1, x_2 \sim p_{\Omega_{obs}}(x_1, x_2) \\ y \sim p_\Omega(y|x_1, x_2)}} \left[ \frac{1 - p(m_1, m_2)}{1 - p(m_1, m_2 | x_1, x_2, y)} \log \left( \frac{p_\Omega(x_1, x_2, y)}{p_\Omega(x_1, x_2) p_\Omega(y)} \right) \right]$$

*Proof.* Let $m = (m_1, m_2)$,

$$I_\Omega(Y : (X_1, X_2))$$

$$= \mathbb{E}_\Omega \left[ \log \left( \frac{p_\Omega(x_1, x_2, y)}{p_\Omega(x_1, x_2) p_\Omega(y)} \right) \right]$$

$$= \mathbb{E}_{\substack{x_1, x_2 \sim p_{\Omega_{obs}}(x_1, x_2) \\ y \sim p_\Omega(y|x_1, x_2)}} \left[ \frac{p_\Omega(x_1, x_2, y)}{p_{\Omega_{obs}}(x_1, x_2, y)} \log \left( \frac{p_\Omega(x_1, x_2, y)}{p_\Omega(x_1, x_2) p_\Omega(y)} \right) \right]$$

$$= \mathbb{E}_{\substack{x_1, x_2 \sim p_{\Omega_{obs}}(x_1, x_2) \\ y \sim p_\Omega(y|x_1, x_2)}} \left[ \frac{p_\Omega(x_1, x_2, y)}{p_\Omega(x_1, x_2, y \mid m = 0)} \log \left( \frac{p_\Omega(x_1, x_2, y)}{p_\Omega(x_1, x_2) p_\Omega(y)} \right) \right]$$

$$= \mathbb{E}_{\substack{x_1, x_2 \sim p_{\Omega_{obs}}(x_1, x_2) \\ y \sim p_\Omega(y|x_1, x_2)}} \left[ \frac{p_\Omega(x_1, x_2, y)}{\frac{p(m=0|x_1, x_2, y) p_\Omega(x_1, x_2, y)}{p(m=0)}} \log \left( \frac{p_\Omega(x_1, x_2, y)}{p_\Omega(x_1, x_2) p_\Omega(y)} \right) \right]$$

$$= \mathbb{E}_{\substack{x_1, x_2 \sim p_{\Omega_{obs}}(x_1, x_2) \\ y \sim p_\Omega(y|x_1, x_2)}} \left[ \frac{1 - p(m = 1)}{1 - p(m = 1|x_1, x_2, y)} \log \left( \frac{p_\Omega(x_1, x_2, y)}{p_\Omega(x_1, x_2) p_\Omega(y)} \right) \right]$$

$$\square$$

That is, to estimate the mutual information under the true data distribution, we adjust for the shift in $p_{\Omega_{obs}}(x_1, x_2) \mapsto p_\Omega(x_1, x_2)$ and sample $y$ from the IPW-adjusted (parametrized approximations) of $p_\Omega(y \mid x_1, x_2)$.

## B PARTIAL INFORMATION DECOMPOSITION (PID)

Partial information decomposition (PID Williams & Beer (2010)) consists in decomposing the total mutual information (McGill, 1954; Te Sun, 1980) between a target variable and two input variables into information about the target variable that both input variables share ("Shared" information), only one input variable has ("Unique" information) and emerges from the interactions of both ("Complementary" information). Bertschinger et al. (2014) introduces bounds for these, reiterated below. In this Appendix, we express these bounds as entropy. First, Table 5 summarizes the notations used in the literature and those used in our work.

**Table 5.** Quantities and associated variables. Note that the four information measures are approximations.

| Quantity | Bertschinger | ICYM$^2$I |
|---|---|---|
| Input Variable 1 | $Y$ | $X_1$ |
| Input Variable 2 | $Z$ | $X_2$ |
| Target Variable | $X$ | $Y$ |
| Redundant / Shared Information | $\widetilde{SI}(X:Y:Z)$ | $\widetilde{SI}(Y:X_1;X_2)$ |
| Unique Information (Input Variable 1) | $\widetilde{UI}(X:Y\backslash Z)$ | $\widetilde{UI}(Y:X_1\backslash X_2)$ |
| Unique Information (Input Variable 2) | $\widetilde{UI}(X:Z\backslash Y)$ | $\widetilde{UI}(Y:X_2\backslash X_1)$ |
| Synergistic / Complementary Information | $\widetilde{CI}(X:Y;Z)$ | $\widetilde{CI}(Y:X_1;X_2)$ |

PID decomposition of the three-way mutual information $I(Y:(X_1,X_2))$ results in the quantities of interest as follows:

$$I(Y:(X_1,X_2)) = \underbrace{SI(Y:X_1;X_2)}_{\text{Shared}} + \underbrace{UI(Y:X_1\backslash X_2)}_{\text{Unique 1}} + \underbrace{UI(Y:X_2\backslash X_1)}_{\text{Unique 2}} + \underbrace{CI(Y:X_1;X_2)}_{\text{Complementary}}$$

Let $\Delta$ be the space of all distributions over $(X_1,X_2,Y)$ and let $\Omega$ denote the true data distribution (without missingness) and define $\Delta_\Omega := \big\{q \in \Delta : q(X_i = x_i, Y = y) = p_\Omega(X_i = x_i, Y = y) \,\forall x_i \in \mathcal{X}_i, y \in \mathcal{Y}, i \in \{1,2\}\big\}$. That is, $\Delta_\Omega$ is the set of all distributions over $(X_1,X_2,Y)$ such that the two-way joints between $X_i$ and $Y$ match the true data-generating distribution. Equipped with this set, Bertschinger et al. (2014) provides the following bounds $\widetilde{SI}$, $\widetilde{UI}$, and $\widetilde{CI}$ on the analogous quantities:

$$
\begin{aligned}
\widetilde{SI}(Y:X_1;X_2) &= \max_{q\in\Delta_\Omega} CoI_q(Y;X_1;X_2) \\
&= \max_{q\in\Delta_\Omega} [I_q(Y:X_1) - I_q(Y:X_1|X_2)] \\
&= \max_{q\in\Delta_\Omega} [[I_q(Y:(X_1,X_2)) - I_q(Y:X_2|X_1)] - I_q(Y:X_1|X_2)] \\
&= \max_{q\in\Delta_\Omega} [I_q(Y:(X_1,X_2)) - [I_q(Y:X_2|X_1) + I_q(Y:X_1|X_2)]] \\
\widetilde{UI}(Y:X_1\backslash X_2) &= \min_{q\in\Delta_\Omega} I_q(Y:X_1|X_2) \\
&= \min_{q\in\Delta_\Omega} [I_q(Y:(X_1,X_2)) - I_q(Y:X_2)] \\
\widetilde{UI}(Y:X_2\backslash X_1) &= \min_{q\in\Delta_\Omega} I_q(Y:X_2|X_1) \\
&= \min_{q\in\Delta_\Omega} [I_q(Y:(X_1,X_2)) - I_q(Y:X_1)] \\
\widetilde{CI}(Y:X_1;X_2) &= I_\Omega(Y:(X_1,X_2)) - \min_{q\in\Delta_\Omega} I_q(Y:(X_1,X_2))
\end{aligned}
$$

In this context, Bertschinger et al. (2014) demonstrates that solving the optimization for $q \in \Delta_\Omega$ that satisfies one of the four conditions above is sufficient to obtain all the quantities of interest.

Importantly, these bounds are tight if there exists a $q_0 \in \Delta_\Omega$ such that $\widetilde{CI}_{q_0}(Y : X_1; X_2) = 0$. Bertschinger et al. (2014) further shows that under common (but unverifiable) assumptions on the data-generating process, the inequalities are tight for all $q \in \Delta_\Omega$. This results in a compelling argument for relying on these entities, as it suggests that it is not possible to decide whether complementary information exists when only the marginals $(Y, X_1)$ and $(Y, X_2)$ are known.

**Formulating PID quantities in terms of entropy.** For stability, we propose to formalize the previous bound in terms of entropy, $H(\cdot)$, defined for general distributions of $X$ and $Y$ as follows:

$$H(X) := -\sum_{x \in \mathcal{X}} p(x) \log p(x)$$

$$H(Y, X) := -\sum_{x \in \mathcal{X}, y \in \mathcal{Y}} p(y, x) \log (p(y, x))$$

$$H(Y|X) := -\sum_{x \in \mathcal{X}, y \in \mathcal{Y}} p(y, x) \log \left( \frac{p(y, x)}{p(x)} \right)$$

$$= H(Y, X) - H(X)$$

Using these notations, the mutual information $I(\cdot)$ can be defined as:

$$I(Y : X) := H(X) - H(X|Y)$$
$$= H(X) + H(Y) - H(Y, X)$$
$$I(Y : X_2|X_1) := H(Y, X_1) + H(X_1, X_2) - H(Y, X_1, X_2) - H(X_1)$$

The previous quantities of interest can then be derived as:

$$I(Y : (X_1, X_2)) := I(Y : X_1) + I(Y : X_2|X_1)$$
$$= \underbrace{H(Y) + \cancel{H(X_1)} - \cancel{H(Y, X_1)}}_{I(Y:X_1)} + \underbrace{\cancel{H(Y, X_1)} + H(X_1, X_2) - H(Y, X_1, X_2) - \cancel{H(X_1)}}_{I(Y:X_2|X_1)}$$
$$= H(Y) + H(X_1, X_2) - H(Y, X_1, X_2)$$

where the first equation comes from the chain rule of mutual information (Wyner, 1978).

Similarly, we can get the expression for co-information $CoI(Y; X_1; X_2)$:

$$CoI(Y; X_1; X_2) = I(Y : X_1) + I(Y : X_2) - I(Y : (X_1, X_2))$$
$$= \underbrace{[H(Y) - H(Y|X_1)]}_{I(Y:X_1)} + \underbrace{[H(Y) - H(Y|X_2)]}_{I(Y:X_2)}$$
$$\quad - \underbrace{[H(Y) + H(X_1, X_2) - H(Y, X_1, X_2)]}_{I(Y:(X_1,X_2))}$$
$$= [H(Y) - [H(Y, X_1) - H(X_1)]] + [\cancel{H(Y)} - [H(Y, X_2) - H(X_2)]]$$
$$\quad - [\cancel{H(Y)} + H(X_1, X_2) - H(Y, X_1, X_2)]$$
$$= H(Y) + H(X_1) + H(X_2)$$
$$\quad - [H(X_1, X_2) + H(Y, X_1) + H(Y, X_2)]$$
$$\quad + H(Y, X_1, X_2)$$

The PID bounds can then be expressed in terms of entropy:

$$
\begin{aligned}
\widetilde{SI}(Y : X_1; X_2) =& \max_{q \in \Delta_\Omega} CoI_q(Y; X_1; X_2) \\
=& \max_{q \in \Delta_\Omega} [H_q(Y) + H_q(X_1) + H_q(X_2) \\
& - [H_q(X_1, X_2) + H_q(Y, X_1) + H_q(Y, X_2)] + H_q(Y, X_1, X_2)] \\
\widetilde{UI}(Y : X_1 \backslash X_2) =& \min_{q \in \Delta_\Omega} I_q(Y : X_1 | X_2) \\
=& \min_{q \in \Delta_\Omega} [H_q(Y, X_2) + H_q(X_1, X_2) - H_q(Y, X_1, X_2) - H_q(X_2)] \\
\widetilde{UI}(Y : X_2 \backslash X_1) =& \min_{q \in \Delta_\Omega} I_q(Y : X_2 | X_1) \\
=& \min_{q \in \Delta_\Omega} [H_q(Y, X_1) + H_q(X_1, X_2) - H_q(Y, X_1, X_2) - H_q(X_1)] \\
\widetilde{CI}(Y : X_1; X_2) =& I_\Omega(Y : (X_1, X_2)) - \min_{q \in \Delta_\Omega} I_q(Y : (X_1, X_2)) \\
=& [H_\Omega(Y) + H_\Omega(X_1, X_2) - H_\Omega(Y, X_1, X_2)] \\
& - \min_{q \in \Delta_\Omega} [H_q(Y) + H_q(X_1, X_2) - H_q(Y, X_1, X_2)]
\end{aligned}
$$

where $H_q(\cdot)$ and $H_\Omega(\cdot)$ is the entropy of a variable under probability distributions $q$ and $\Omega$, respectively.

## C   ICYM$^2$I

### C.1   ICYM$^2$I-LEARN

Table 6 summarizes the proposed approach to estimate performance using $\Omega_{\text{obs}}$. Critically, one must correct both training and evaluation to obtain the performance on $\Omega$.

**Table 6.** ICYM$^2$I: Inverse probability weighting-adjusted multimodal training and evaluation under missingness shift.

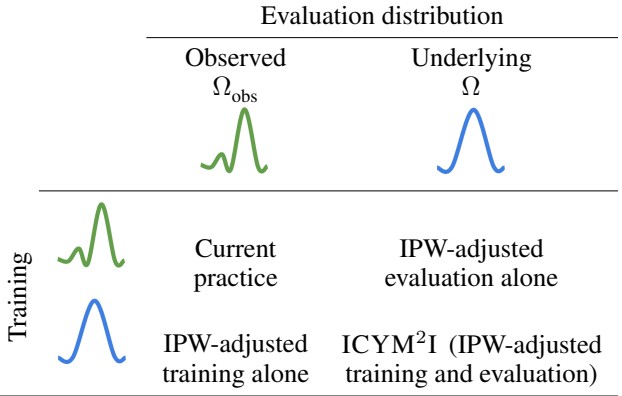

|  | Evaluation distribution | |
|---|---|---|
|  | Observed $\Omega_{\text{obs}}$ | Underlying $\Omega$ |
| Training | Current practice | IPW-adjusted evaluation alone |
|  | IPW-adjusted training alone | ICYM$^2$I (IPW-adjusted training and evaluation) |

### C.2   ICYM$^2$I-PID

The following describes the algorithm to estimate PID under missingness.

---

**Algorithm 1** ICYM$^2$I-PID

---

**Require:** $X_1, X_2, Y \sim p_{\Omega_{\text{obs}}}$
  1:  # Step 1: Adjust $\Omega_{\text{obs}} \mapsto \Omega$.
  2:  Estimate missingness mechanisms $p_{\Omega_\phi}(M_1, M_2, M_Y \mid C)$ for IPW.
  3:  # Step 2: Train corrected unimodal and multimodal models.
  4:  Training each model with weighting IPW-loss: $f(y \mid x_i) \approx p_\Omega(y \mid x_i), \forall i \in \{1, 2\}$, and $f(y \mid x_1, x_2) \approx p_\Omega(y \mid x_1, x_2)$.
  5:  # Step 3: PID optimization.
  6:  Initialize parameterizations $\theta$ for $q$: $f_i(y \mid x_i), \forall i \in \{1, 2\}$.
  7:  $q_\theta(y \mid x_1, x_2) \leftarrow \exp\left(f_1(y \mid x_1) f_2(y \mid x_2)^T\right)$
  8:  **while** not converged **do**
  9:      **for** samples in batch **do**
 10:          # Ensure $q \in \Delta_\Omega^{\text{ICYM}^2\text{I}}$ by projection.
 11:          $q_\theta(y \mid x_1, x_2) \leftarrow \texttt{SINKHORN-KNOPP}(q_\theta(y \mid x_1, x_2), \{p_\Omega(y, x_i)\}_{i=1}^2)$.
 12:          Estimate the loss $I_q(Y : (X_1, X_2))$ as a batch sample mean.
 13:          $\theta \leftarrow \theta - \nabla_\theta I_q(Y : (X_1, X_2))$.
 14:      **end for**
 15:  **end while**
      # Step 4: Estimate mutual information under $p_\Omega$.
 16:  Estimate $I_\Omega(Y : (X_1, X_2))$, and $I_\Omega(Y : X_i), \forall i \in \{1, 2\}$ using adjustment in Appendix A.
 17:  $PID(\Omega) \leftarrow (\widetilde{CI}(Y : X_1; X_2), \widetilde{SI}(Y : X_1; X_2), \widetilde{UI}(Y : X_1 \backslash X_2), \widetilde{UI}(Y : X_2 \backslash X_1))$
 18:  **return** $PID(\Omega)$

---

The traditional `SINKHORN-KNOPP` algorithm updates a matrix to enforce its marginals to be unit vectors. In our work, we adapt the algorithm to enforce the marginals to match $p_\Omega$-marginals, ensuring that $q_\theta(\cdot) \in \Delta_\Omega$. To ensure proper gradient propagation and reduce memory use, we use the unrolled `SINKHORN-KNOPP` (Sinkhorn & Knopp, 1967; Cuturi, 2013) algorithm. In the following, we use subscripts $q_{x_1, x_2}$ to denote $q_\theta(y, x_1, x_2)$ and $p_{x_i}$ to denote $p_\phi(y, x_i)$. The algorithm is detailed below:

---

**Algorithm 2** Unrolled `SINKHORN-KNOPP` update

---

**Require:** $q_{x_1 x_2}, p_{x_1}, p_{x_2}$, tolerance `atol`
1: $q_{x_1} \leftarrow \sum_{x_2} q_{x_1 x_2}$
2: $q_{x_2} \leftarrow \sum_{x_1} q_{x_1 x_2}$
3: **while do**
4:     # Avoid update if both exit conditions have been met.
5:     **if** $\left| \frac{q_{x_1} - p_{x_1}}{p_{x_1}} \right| \leq$ `atol` **and** $\left| \frac{q_{x_2} - p_{x_2}}{p_{x_2}} \right| \leq$ `atol` **then**
6:         **return** $q_{x_1 x_2}$
7:     **end if**
8:     # Update marginal.
9:     $q_{x_1 x_2} \leftarrow \frac{q_{x_1 x_2}}{q_{x_2}} \cdot p_{x_2}$
10:     $q_{x_1} \leftarrow \sum_{x_2} q_{x_1 x_2}$
11:     # If the other marginal still matches, done.
12:     **if** $\left| \frac{q_{x_1} - p_{x_1}}{p_{x_1}} \right| \leq$ `atol` **then**
13:         **return** $q_{x_1 x_2}$
14:     **end if**
15:     # Repeat for the other marginal.
16:     $q_{x_1 x_2} \leftarrow \frac{q_{x_1 x_2}}{q_{x_1}} \cdot p_{x_1}$
17:     $q_{x_2} \leftarrow \sum_{x_1} q_{x_1 x_2}$
18:     **if** $\left| \frac{q_{x_2} - p_{x_2}}{p_{x_2}} \right| \leq$ `atol` **then**
19:         **return** $q_{x_1 x_2}$
20:     **end if**
21: **end while**

---

# D    BIT-WISE LOGITS

In this section, we perform a sensitivity analysis of the logit setting presented in Section 4.1 under two additional missingness patterns: MCAR (Missing Completely at Random) and MNAR (Missing Not at Random). In this setting, we fit a logistic regression to estimate the probability of missingness on the observed modality, which is then used to estimate the IPW. The performance estimates and PID for these two missingness processes are illustrated in Tables 7 and 8.

Since MCAR does not result in a distribution shift, one expects the same performance estimates for both the full and observed populations. Furthermore, in this setting, the IPW correction corresponds to a constant value, as any point has the same probability of observing both modalities. This correction also results in no change in performance estimates.

On the contrary, MNAR patterns do not guarantee similar behavior. Particularly, this missingness process may result in a distribution shift that cannot be assessed or accounted for without assumptions about the data distribution, as one does not observe the covariates that impact the missingness process. The results demonstrate that both the observed and corrected strategies result in biased estimates.

**Table 7.** Impact of missingness on multimodality information for bitwise logic operators under MCAR. Parentheses denote standard deviation across batches.

| | | AUROC | | | Information Decomposition | | | |
|---|---|---|---|---|---|---|---|---|
| | | $X_1$ | $X_2$ | $X_1 + X_2$ | Unique 1 | Unique 2 | Shared | Complementary |
| AND | Oracle | 0.83 (0.01) | 0.84 (0.01) | 1.00 (0.00) | 0.05 (0.00) | 0.03 (0.00) | 0.26 (0.00) | 0.47 (0.00) |
| | Observed | 0.83 (0.01) | 0.83 (0.01) | 1.00 (0.00) | 0.05 (0.00) | 0.03 (0.00) | 0.23 (0.00) | 0.52 (0.00) |
| | ICYM$^2$I | 0.83 (0.01) | 0.85 (0.01) | 1.00 (0.00) | 0.03 (0.00) | 0.06 (0.00) | 0.27 (0.00) | 0.44 (0.00) |
| OR | Oracle | 0.84 (0.01) | 0.83 (0.01) | 1.00 (0.00) | 0.04 (0.00) | 0.05 (0.00) | 0.27 (0.00) | 0.46 (0.00) |
| | Observed | 0.84 (0.01) | 0.84 (0.01) | 1.00 (0.00) | 0.06 (0.00) | 0.03 (0.00) | 0.25 (0.00) | 0.51 (0.00) |
| | ICYM$^2$I | 0.85 (0.01) | 0.83 (0.01) | 1.00 (0.00) | 0.06 (0.00) | 0.02 (0.00) | 0.25 (0.00) | 0.51 (0.00) |
| XOR | Oracle | 0.51 (0.02) | 0.49 (0.01) | 1.00 (0.00) | 0.00 (0.00) | 0.00 (0.00) | 0.00 (0.00) | 0.99 (0.00) |
| | Observed | 0.51 (0.02) | 0.50 (0.02) | 1.00 (0.00) | 0.00 (0.00) | 0.00 (0.00) | 0.00 (0.00) | 0.95 (0.00) |
| | ICYM$^2$I | 0.51 (0.02) | 0.51 (0.02) | 1.00 (0.00) | 0.00 (0.00) | 0.00 (0.00) | 0.00 (0.00) | 0.95 (0.00) |

**Table 8.** Impact of missingness on multimodality information for bitwise logic operators under MNAR. Parentheses denote standard deviation across batches.

| | | AUROC | | | Information Decomposition | | | |
|---|---|---|---|---|---|---|---|---|
| | | $X_1$ | $X_2$ | $X_1 + X_2$ | Unique 1 | Unique 2 | Shared | Complementary |
| AND | Oracle | 0.83 (0.01) | 0.84 (0.01) | 1.00 (0.00) | 0.05 (0.00) | 0.03 (0.00) | 0.26 (0.00) | 0.47 (0.00) |
| | Observed | 0.93 (0.01) | 0.67 (0.01) | 1.00 (0.00) | 0.45 (0.00) | 0.00 (0.00) | 0.17 (0.00) | 0.33 (0.00) |
| | ICYM$^2$I | 0.93 (0.01) | 0.67 (0.01) | 1.00 (0.00) | 0.45 (0.00) | 0.00 (0.00) | 0.17 (0.00) | 0.33 (0.00) |
| OR | Oracle | 0.84 (0.01) | 0.83 (0.01) | 1.00 (0.00) | 0.04 (0.00) | 0.05 (0.00) | 0.27 (0.00) | 0.46 (0.00) |
| | Observed | 0.78 (0.01) | 0.95 (0.01) | 1.00 (0.00) | 0.00 (0.00) | 0.17 (0.00) | 0.11 (0.00) | 0.23 (0.00) |
| | ICYM$^2$I | 0.78 (0.01) | 0.95 (0.01) | 1.00 (0.00) | 0.00 (0.00) | 0.17 (0.00) | 0.11 (0.00) | 0.23 (0.00) |
| XOR | Oracle | 0.51 (0.02) | 0.49 (0.01) | 1.00 (0.00) | 0.00 (0.00) | 0.00 (0.00) | 0.00 (0.00) | 0.99 (0.00) |
| | Observed | 0.80 (0.02) | 0.52 (0.02) | 1.00 (0.00) | 0.35 (0.00) | 0.07 (0.00) | 0.00 (0.00) | 0.61 (0.00) |
| | ICYM$^2$I | 0.80 (0.02) | 0.52 (0.02) | 1.00 (0.00) | 0.35 (0.00) | 0.07 (0.00) | 0.00 (0.00) | 0.61 (0.00) |

# E  SYNTHETIC DATA RESULTS

**Data generation.** Our work builds on the example introduced in (Liang et al., 2024a), in which we enforce additional missingness. Three latent variables ($z_1$, $z_2$, and $z_c$) are drawn from multi-dimensional clustered data; the observed covariates are a concatenation of $z_c$ and one of the other latent variables, as illustrated in Figure 3. Then, the outcome $Y$ is generated as $Y = \sigma(p_1 \mathbb{E}(z_1) + p_2 \mathbb{E}(z_2) + (1 - p_1 - p_2)\mathbb{E}(z_c))$, with the proportion $p_i \in [0, 1]$ such that $p_1 + p_2 \le 1$. We simulate datasets with varying values of $p_1$ and $p_2$. Then, we enforce a $50\%$ MAR missingness pattern in $X_2$ by modeling the probability of missingness. We do this by clustering $X_1$ into 100 groups using Kmeans. Then, the probability of missingness is generated using a random forest that regresses $X_1$ to predict $c_j \cdot Y$.

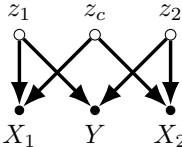

**Figure 3.** Data generating processes for synthetic experiments. $z_i$ denote latent vectors, while all other variables are observed. Filled point nodes are observed variables, while unfilled nodes are unobserved.

**Empirical setting.** Data were split into three: 80% for training, 10% for validation, and the rest for testing. We consider neural networks with 2 hidden layers with 32 nodes, trained using an Adam optimizer (Kingma & Ba, 2014) with a learning rate of 0.001 over 100 epochs. Our evaluation relies on discriminative performance measured through AUROC.

**Estimating predictive performance under $\Omega_{\mathbf{obs}}$.** Table 9 presents the PID obtained under different corrections. These results underline the importance of correcting both training and evaluation, as proposed in ICYM²I, to best align with the PID one would obtain on $\Omega$, as shown by the smallest Root Mean Squared Error (RMSE) observed when both corrections are applied. Note that in this setting, we rely on the true IPW correction that one would obtain with a properly specified model, as the MAR setting is met.

**Table 9.** Comparison between estimated PID using training and PID corrections, and oracle PID on $\Omega$. $\epsilon$ denotes the RMSE between estimated and oracle PIDs.

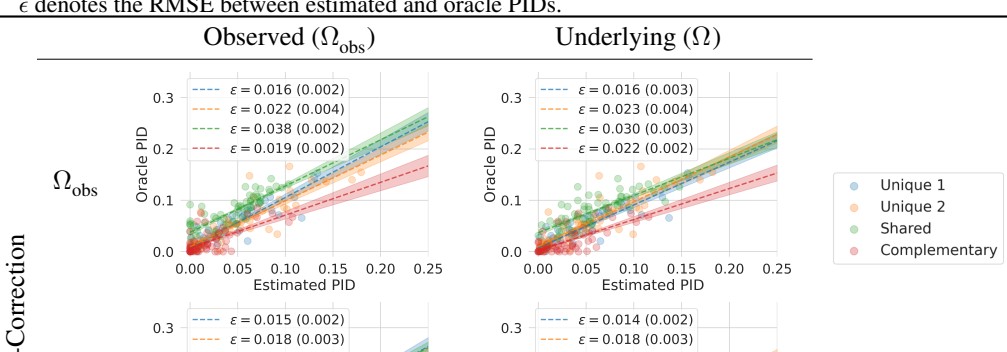

# F  SEMI-SYNTHETIC DATA RESULTS

## F.1  UR-FUNNY

We illustrate the impact of missingness on estimating the informativeness of different modalities on real-world data with UR-FUNNY (Hasan et al., 2019), a multimodal dataset for humor detection from human speech used in affective computing. The dataset comprises text, audio, and visual modalities from 10 - 20 second videos sourced from TED talks, and the task is to detect whether a punchline would trigger a laugh. Labels were generated using the markup "(Laughter)" (Chen & Lee, 2017) from the transcript.

**Dataset.** The processed dataset from MultiBench (Liang et al., 2021) is a modality-complete dataset with 10,166 samples of paired audio, text, and vision embeddings. Audio embeddings were generated with COVAREP (Degottex et al., 2014), text with Glove (Pennington et al., 2014), and visual features through the Facet (Yuan et al., 2008) library and OpenFace (Baltrušaitis et al., 2016), and aligned using the Penn Phonetics Lab Forced Aligner (P2FA) (Yuan et al., 2008).

### F.1.1  MISSING AT RANDOM (MAR)

**Enforcing missingness.** To explore the impact of missingness on informativeness, we simulate a MAR missingness pattern on the audio and visual features given the textual modality. We vary the missingness from 30% to 70%, using the same mechanism as described for synthetic data. This semi-synthetic setting enables the evaluation of the proposed correction as the missingness mechanism is known. Note that the original dataset does not contain missing values, as the source data (TED Talks) have transcripts, and data labeling was generated based on these transcripts. However, settings with systematic transcripts are rare and may reflect a shift from the audio and textual modalities observed online for which such a match may not exist.

**Results.** Following the same empirical setting as in the synthetic experiment for each missingness rate, we measure the impact of missingness on PID decomposition. Figure 4 displays the PID values obtained under three strategies:

- Observed: All quantities are estimated using $\Omega_{\text{obs}}$.
- ICYM$^2$I: All quantities are estimated using $\Omega_{\text{obs}}$ but corrected for the distribution shift through IPW.
- Oracle: All quantities are estimated on $\Omega$.

This figure shows that the proposed strategy is consistently closer to the Oracle's PID values. This demonstrates that under Assumption A, the proposed correction yields better estimates of each modality's informativeness – specifically, the audio-visual modality (Unique 1) carries more information.

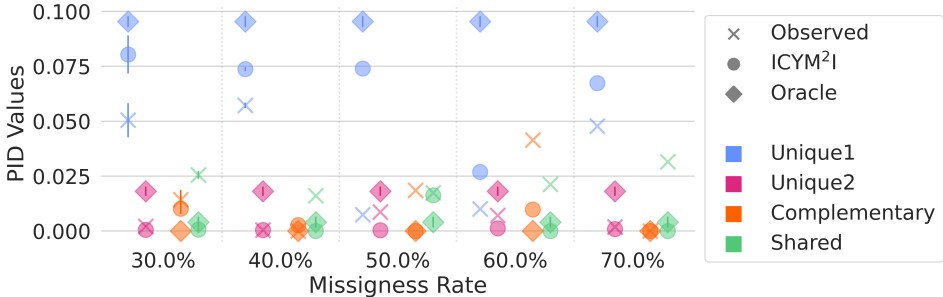

**Figure 4:** Comparison between estimated PID values under increasing missingness in UR-FUNNY.

### F.1.2  MISSING NOT AT RANDOM (MNAR)

**Enforcing missingness.** A central assumption of our method is that missingness is MAR. We propose to analyze the impact of violations of this assumption, specifically the presence of MNAR

patterns, on the quality of estimates obtained using our correction. To this end, we simulate audio and visual missingness as a function of the modality itself. Similarly to the previous analysis, we vary missingness from 30% to 70%. To estimate propensity in this setting, we rely on a logistic regression model based on the fully observed modality.

**Results.** Figure 5 shows the PID values obtained under the three previously described strategies. Critically, the proposed correction leads to performance similar to the model without correction, as the missingness probabilities cannot be estimated from the observed modality. This example illustrates the importance of assessing the plausibility of Assumption B in real-world settings, as no theoretical guarantees hold in such settings.

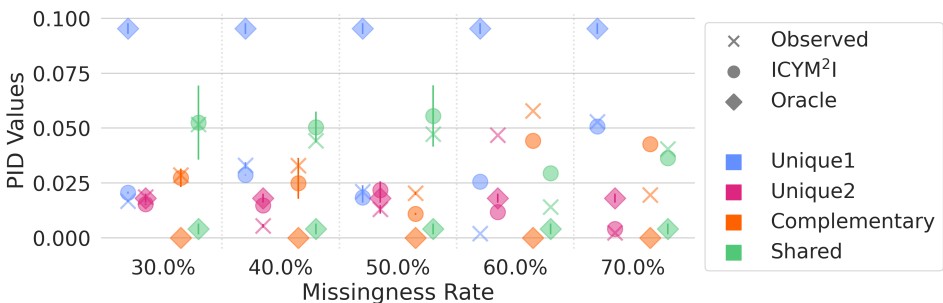

**Figure 5:** Comparison between estimated PID values under increasing missingness in UR-FUNNY.

## F.2 HATEFUL MEMES

We run experiments using the dataset from the Hateful Memes Challenge (Kiela et al., 2020), which investigates text-image multimodal reasoning in the context of hate speech detection in online memes. The dataset comprises text-image pairs with an associated label indicating hate speech.

**Dataset.** We utilize the Kaggle version of the Facebook Hateful Memes dataset, as referenced in the Holistic Evaluation of Multimodal Foundation Models (HEMM) (Liang et al., 2024b) repository. Our analysis focuses on the 9,000 samples with associated labels. For each sample, embeddings were extracted for both modalities using a ResNet-50 (He et al., 2016) for images and a BERT-base-uncased (Devlin et al., 2019) model for text. The proposed ResNet-50 was pretrained on ImageNet (Deng et al., 2009) with the final layer replaced to extract 2048-dimensional feature vectors, and BERT-base-uncased (Devlin et al., 2019) extracts embeddings of dimension 784 from the penultimate layer.

**Enforcing missingness.** Similarly to the previous experiment, we vary the missingness from 30% to 70% by enforcing the same MAR missingness mechanism on the text modality, given the image modality, as we assume not all memes may contain text. Note that memes in the dataset were created by combining text from collected online memes with images sourced from stock images on Getty Images. Consequently, the dataset did not contain missing modality, but may not match the true distribution of memes one would observe online.

**Results.** As above, we measure the impact of increasing percentages of missingness on PID estimates. While the missingness mechanism results in a limited distribution shift, and therefore small differences in estimates between the corrected and observed strategies, the difference at 70% missingness shows the superiority of the proposed methodology in recovering the Unique contributions.

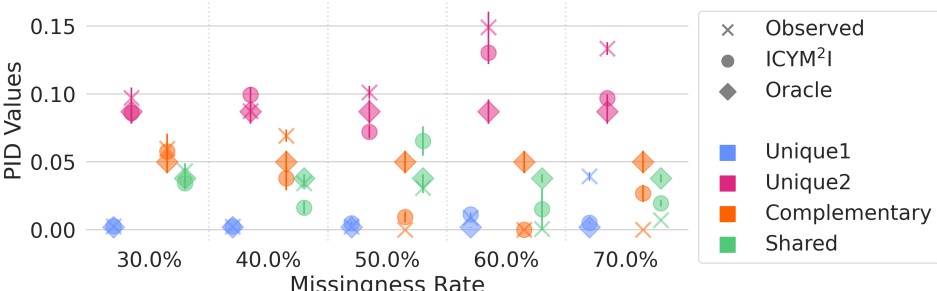

**Figure 6.** Comparison between estimated PID values under increasing missingness in Hateful Memes.

# G  STRUCTURAL HEART DISEASE DATA PROCESSING

## G.1  EMBEDDING GENERATION

We generate embeddings using modality-specific foundation models–ECG embeddings are generated using ECG-FM (McKeen et al., 2024) and CXR embeddings with ELIXR-C (Xu et al., 2023).

All electrocardiograms were 10-second, standard 12-lead ECG signals collected and abstracted to 250 Hz, which we resampled to 500 Hz, and standard normalized by channel to match the inputs for ECG-FM (McKeen et al., 2024). We used the version of ECG-FM with weights pretrained on MIMIC-IV (Johnson et al., 2023; Goldberger et al., 2000) and PhysioNet 2021 (Reyna et al., 2021; 2022). We averaged the output feature embeddings along the temporal dimension and flattened them to produce vectors of length 768.

The chest radiographs used in our study were all postero-anterior (PA) view CXRs. We extracted pixel values from the DICOM files as grayscale images, center-cropped each image along the shorter dimension, applied contrast-limited adaptive histogram equalization (CLAHE) (Pizer et al., 1987) with a clip limit of 0.2, and resized each image to $1284 \times 1284$ pixels. All outputted embeddings were flattened to 4098-dimensional vectors.

## G.2  IPW CORRECTION

To address missingness in the observed CXRs, we apply the proposed propensity-based correction. The propensity scores are obtained from a logistic regression model using the ECG embedding, along with sex and age as predictors, serving as proxies for the socio-medical factors that influence whether a CXR is collected. Controlling for these covariates aims to render the MAR assumption more plausible. In practice, all relevant covariates, even outside of modalities being modeled, can be used for the correction.

# H  COMPUTE INFRASTRUCTURE

All experiments were performed on a server with an AMD EPYC 7313 CPU, 256 GB of memory, and two NVIDIA RTX A6000 GPUs, as well as a server with an Intel Xeon E5-2640 CPU, 128 GB of memory, and a NVIDIA GTX Titan X GPU. Our software stack includes Python 3.12, PyTorch 2.2.1 (Paszke et al., 2019), and standard Python scientific libraries. Chest radiograph embeddings used Tensorflow 2.19 (Abadi et al., 2015) and Tensorflow-Text 2.19 based on the requirements for the ELIXR models (Xu et al., 2023). Electrocardiogram embeddings were generated using an environment with Python 3.9 and fairseq-signals 1.0 to match the requirements for fairseq-signals and ECG-FM (McKeen et al., 2024). Generating embeddings for our structural heart disease data took approximately 10 hours on our server with a Titan X GPU. All synthetic experiments require 12 hours of compute time using one NVIDIA RTX A6000 GPU.

