# OpenReview forum: "ICYM$^2$I: The illusion of multimodal informativeness under missingness"
_ICLR.cc/2026/Conference — ICLR 2026 Poster_

### Official Review · Reviewer_pctf · 2025-10-21

**Soundness:** 3
**Presentation:** 2
**Contribution:** 3
**Rating:** 4
**Confidence:** 2

**Summary:**

This paper addresses missingness-induced distribution shifts in multimodal learning, a largely overlooked issue where source and target environments differ in observed modalities. It notes that naive estimation without accounting for this shift biases modality informativeness and predictive utility estimates. To solve this, it proposes the $ICYM^2I$ framework, which uses inverse probability weighting correction under the MAR assumption to evaluate performance and information gain under missingness. The framework is validated on synthetic, semi-synthetic, and real-world datasets to demonstrate its utility.

**Strengths:**

1. Framing missingness in multimodal learning as an intrinsic distribution shift between source and target environments is novel, as well as formalizing its mechanisms (MCAR, MAR, and MNAR) and showing unaddressed shift biases modality informativeness and predictive utility estimates.
2. This paper proposes a double inverse probability weighting (IPW) framework under the realistic MAR assumption to correct both training and evaluation, enabling unbiased assessment of predictive performance and information gain under missingness.
3. The proposed method integrates IPW into Partial Information Decomposition (PID) to adjust for $\Omega_{obs} \mapsto \Omega$ shift, designing an autodifferentiable algorithm with modified Sinkhorn-Knopp to handle high-dimensional data for unbiased information decomposition
4. This paper applies the framework to structural heart disease detection, revealing chest radiographs have minimal independent informativeness, bridging methodological advances with real-world healthcare.

**Weaknesses:**

1. The paper relies heavily on the Missing At Random (MAR) assumption, but real-world multimodal scenarios often involve Missing Not At Random (MNAR). It provides no solutions for MNAR and only briefly mentions its limitations, lacking sensitivity analysis for MAR violations. The authors could add experiments on MNAR with plausible assumptions (e.g., simulating unobserved covariates) or explore semi-parametric methods to relax MAR.
2. The PID in $ICYM^2I$ focuses on two modalities, but real multimodal tasks involve more, i.e., 3+ modalities. This paper only mentions a "one-vs-all" approach without detailed implementation or validation, leaving scalability unclear.
3. Experiments simulate MAR using existing modalities (e.g., X1 predicts X2 missing) but ignore real drivers of missingness. For example, in healthcare, CXR missingness may relate to patient insurance status, which is unrelated to existing modalities.
4. The SHD case study uses data from a single academic hospital system, with no validation on external multi-center datasets. This limits the clinical generalization of conclusions.

**Questions:**

none

---

> ### Author Response · Authors · 2025-11-21
>
> Thank you for your thoughtful and constructive feedback. We are glad to see you share our view that formalizing missingness in multimodal learning is a critical and overlooked research challenge. We also appreciate your positive evaluation of the formalization of missingness in multimodal learning as a distribution shift problem, the methodological rigor in our work, and the applicability of our method to real-world multimodal data. Below, we address the concerns raised and provide additional clarifications.
>
> **The paper relies heavily on the Missing At Random (MAR) assumption, but real-world multimodal scenarios often involve Missing Not At Random (MNAR). It provides no solutions for MNAR and only briefly mentions its limitations, lacking sensitivity analysis for MAR violations. The authors could add experiments on MNAR with plausible assumptions (e.g., simulating unobserved covariates) or explore semi-parametric methods to relax MAR.**
>
> Thank you for highlighting the importance of the missing at random (MAR) and positivity assumptions. As you noted, violations of these assumptions may lead to biased performance and information-gain estimates. A core contribution of our work is the formalization of the problem of missingness in multimodal settings. Through this formalization, we highlight the assumptions under which one can address the biases associated with missingness. We believe that transparency in outlining these assumptions is critical to improving the applicability of these strategies and a limitation of methods that often overlook the complexity and type of multimodal missingness.
>
> Further, Appendix D includes a sensitivity analysis with simulations for both the missing completely at random (MCAR) and missing not at random (MNAR) settings. In the MCAR setting, all methodologies yield comparable performance estimates, demonstrating the stability of our method under both MAR and MCAR settings. However, when the missingness of a variable is drawn as a function of the variable itself (MNAR), we observed biased performance and information estimates. Under such missingness patterns, the correction is not guaranteed to eliminate the bias in estimates because the missingness process cannot be modelled from observed covariates, and therefore cannot be corrected.
>
> Regarding performing a sensitivity analysis, we have included an additional experiment (Appendix F.1.2) on the UR-FUNNY real-world dataset, in which missingness is generated based on the missing modality itself. This MNAR setting echoes the results of the logic experiment, in which failing to meet the MAR assumption leads to biased estimates.
>
> To further address your concern, we have clarified the limitations of MAR/MNAR identifiability in the conclusion.
>
> *“The key assumption in our work is that missingness is MAR. No theoretical guarantees exist under MNAR patterns. While distinguishing these assumptions is empirically untestable, practitioners should ensure that this assumption is appropriate for their data. Importantly, MAR is less restrictive than the implicit MCAR assumption made in the multimodal literature, and does not require unrealistic distributional assumptions that one must assume to tackle MNAR patterns.”*
>
> [1] Nabi, Razieh, and Rohit Bhattacharya. "On testability and goodness of fit tests in missing data models." *Uncertainty in Artificial Intelligence*. PMLR, 2023.
> [2] Ji, Feng, Sophia Rabe-Hesketh, and Anders Skrondal. "Diagnosing and handling common violations of missing at random." *psychometrika* 88.4 (2023): 1123-1143.

---

> > ### Author Response · Authors · 2025-11-21
> >
> > **The PID in focuses on two modalities, but real multimodal tasks involve more, i.e., 3+ modalities. This paper only mentions a "one-vs-all" approach without detailed implementation or validation, leaving scalability unclear.**
> >
> > Thank you for this suggestion. We agree that different application domains may involve more than two modalities and appreciate the opportunity to discuss how our work can be extended to these settings. We note that  ICYM^2I contains two components.
> >
> > First, note that ICYM^2I-learn, our method for performance estimation (Section 4.2), can natively take in an arbitrary number of input modalities, as long as practitioners ensure that MAR assumptions are valid.
> >
> > Second, ICYM^2I-PID (Section 4.3), the information gain decomposition in our work, relies on the Partial Information Decomposition (PID) bounds introduced by [1]. These bounds only cover interactions between two input variables and a target variable. While other work has proposed multivariate extensions to PID, there is no consensus on how to attribute the information gain to each modality [2, 3].
> >
> > Given this, decomposing interactions between more than two input modalities in a one-to-one manner remains a theoretically open challenge. A one-versus-rest approach to ICYM^2I-PID is a straightforward implementation strategy, where a practitioner groups all modalities but one into one and computes the PID decompositions of this grouped modality against the one remaining. This strategy allows one to quantify the theoretical informativeness of each modality independently. However, this strategy does not scale to a large number of modalities and would introduce issues beyond the scope of this paper, which we will explore in future work.
> >
> > To address your concern, we further discuss these possible extensions in the future work section of the updated manuscript.
> >
> > *“Additionally, our work is based on Partial Information Decomposition (PID), which focuses on two input modalities. In practice, practitioners could consider a one-vs-all approach to inform modality informativeness using our method. However, extending the decomposition to more than two modalities remains a theoretical open challenge where notions of mutual information itself are not well outlined beyond three-way mutual information.”*
> >
> > [1] Bertschinger et al. "Quantifying unique information." *Entropy* (2014).
> > [2] Kolchinsky. “A novel approach to the partial information decomposition." *Entropy* (2022).
> > [3] Griffith et al. "Quantifying synergistic mutual information." *Guided self-organization: inception* (2014).
> >
> > **Experiments simulate MAR using existing modalities (e.g., X1 predicts X2 missing) but ignore real drivers of missingness. For example, in healthcare, CXR missingness may relate to patient insurance status, which is unrelated to existing modalities.**
> >
> > Thank you for raising this important point. We agree that the propensity model must control for all covariates C that ensure that the MAR assumption is met. In our case study, we control for sex and age in addition to ECG to capture socio-medical factors that may influence the observation of CXR. Our framework is flexible and can handle covariates to learn propensity scores that may not include existing modalities we’re modeling. To address your concern, we have detailed how the propensity model was obtained in Appendix G.2:
> >
> > *“To address missingness in the observed CXRs, we apply the proposed propensity-based correction. The propensity scores are obtained from a logistic regression model using the ECG embedding, along with sex and age as predictors, serving as proxies for the socio-medical factors that influence whether a CXR is collected. Controlling for these covariates aims to render the MAR assumption more plausible.  In practice, all relevant covariates, even outside of modalities being modeled, can be used for the correction.”*

---

> ### Author Response · Authors · 2025-11-21
>
> **The SHD case study uses data from a single academic hospital system, with no validation on external multi-center datasets. This limits the clinical generalization of conclusions.**
>
> Thank you for raising this important problem. We note that our data includes multiple hospitals and settings (inpatient/outpatient). More crucially, while we agree that external validity would be an important finding, our goal is not to claim clinically generalizable insights on structural heart disease (SHD), but rather to illustrate the application of the proposed methodology in a setting where missingness is meaningful. In real-world settings where data collection is imperfect, one must account for missingness to estimate the informativeness of each modality and potentially their future collection. The promising initial results we have obtained using our method have encouraged us to study this problem using multi-center data, which we are actively pursuing, and to focus on its clinical impact. In this manuscript, we focus on the methodological innovation to test the validity of our proof-of-concept in real-world data, which is more in scope for ICLR. This paper aims to emphasise the methodological contributions. We have clarified this in the case study:
>
> *“While our core contribution is methodological, this section illustrates how ignoring missingness can lead to biased estimates of the informativeness of a given modality in a real-world setting where modalities are commonly missing. Specifically, we study structural heart disease (SHD), a set of conditions that affect the heart's physiology, which is typically diagnosed using transthoracic echocardiograms (TTEs).”*
>
> **Ethics clarification.**
>
> We thank the reviewer for carefully evaluating the ethical aspects of our work. We confirm that all clinical data used in our study were fully de-identified before analysis and obtained under approval of the Institutional Review Board of the clinical center, as outlined in our Ethics Statement (Page 10). Our clinical collaborator is an author of the paper and we have extracted all data in HIPAA-compliant servers; our experiments are also conducted on HIPAA-compliant compute despite being deidentified for extra caution. No protected attributes were used in our analysis, and all data were retrospective, observational data. Our work does not involve any clinical decisions, diagnoses, or interventions. We will further clarify these points in the final manuscript.
>
> We hope these proposed changes address your concerns and that you will consider increasing your score.

---

### Official Review · Reviewer_2rF7 · 2025-10-25

**Soundness:** 3
**Presentation:** 3
**Contribution:** 3
**Rating:** 6
**Confidence:** 1

**Summary:**

I'm not an expert in this area.

**Strengths:**

I'm not an expert in this area.

**Weaknesses:**

I'm not an expert in this area.

**Questions:**

No question.

---

> ### Author Response · Authors · 2025-11-21
>
> We would like to thank the reviewer for considering our submission. We hope that our revisions and clarifications provided in responses to the other reviewers help convey the significance and technical correctness of our contributions.

---

### Official Review · Reviewer_VFgX · 2025-10-31

**Soundness:** 3
**Presentation:** 4
**Contribution:** 4
**Rating:** 6
**Confidence:** 4

**Summary:**

This paper introduces a framework to correct for bias in multimodal learning arising from missing modalities. The authors argue that the informativeness and predictive value of a modality are often misestimated when the missingness process differs between the source and target environments, a realistic but underexplored issue. ICYM2I leverages IPW to adjust both model training and evaluation under the MAR assumption. The method also extends to information-theoretic decomposition, allowing unbiased estimation of modality informativeness through a corrected version of PID. Experiments on synthetic and semi-synthetic datasets demonstrate that ignoring missingness can substantially bias conclusions about modality utility, while ICYM2I provides more accurate informativeness estimates.

**Strengths:**

The paper is conceptually strong and addresses a highly relevant but largely overlooked problem in multimodal learning: the impact of missing modalities on the estimation of informativeness and performance. It offers a clear and rigorous formalization of missingness as a type of distribution shift, which is an important contribution to the field. The proposed use of inverse probability weighting is theoretically grounded in classical missing-data and statistical literature, and its adaptation to multimodal settings is elegant. The integration with information-theoretic concepts, particularly the extension of PID under missingness, is well-motivated and contributes to a deeper understanding of modality interactions. Overall, the paper is well-written, the motivation is convincing, and the theoretical framing is solid and potentially impactful.

**Weaknesses:**

Despite its strong conceptual foundation, the paper’s methodological presentation is somewhat difficult to follow. Understanding the full framework requires frequent reference to the appendix, which makes it challenging to reconstruct the complete algorithmic pipeline from the main text. In particular, Step 2 and Step 10 of Algorithm 1, which concern the estimation of the missingness mechanism $p_{\Omega_\phi}(m|C)$ and its integration into the weighting scheme, are crucial for comprehension and should be explicitly detailed in the main body. It also remains unclear how exactly this missingness model is parameterized or estimated in practice. Moreover, the objective of minimizing $I(Y; (X_1, X_2))$ appears counterintuitive and should be clarified, as mutual information is typically maximized in multimodal fusion. From an empirical perspective, while the synthetic and semi-synthetic experiments are carefully designed and demonstrate the motivation, the work lacks convincing real-world evidence that the proposed correction improves downstream predictive performance. Correcting modality informativeness estimates is valuable, but readers are left wondering whether a practitioner facing genuinely missing modalities would actually see performance gains by applying ICYM2I. Finally, the framework’s dependence on a particular parameterization $q_\theta = \exp(f_1 f_2)$ is restrictive; demonstrating that the approach generalizes across different fusion architectures would strengthen the empirical case and enhance credibility. Lastly, a weakness regarding the submission is that the code has not been shared, either in the supplementary material or via an anonymized GitHub repository, which limits reproducibility and verification of the claims. Lastly, a weakness regarding the submission is that the code has not been shared, either in the supplementary material or via an anonymized GitHub repository, which limits how much reviewers can verify and fully assess the implementation and results.

**Questions:**

Could you elaborate on how  $p_{\Omega_\phi}(m|C)$  is actually modeled and estimated? How sensitive is performance to misspecification of this model?

Have you considered cases with unpaired modalities, where instances across modalities are not aligned? Could you comment how the method could be extended to such cases?

Would the method still hold for other forms of fusion beyond $q_\theta = \exp(f_1 f_2)$ , e.g. attention-based models especially when large in parameter number?

Could one imagine a generative formulation where C and Y are governed by a single latent variable, and would this affect how modality informativeness is estimated?

---

> ### Author Response · Authors · 2025-11-21
>
> Thank you for the positive assessment. We appreciate your recognition of the paper’s conceptual contribution, the formal treatment of missingness, the grounding of our approach in established statistical theory, and the value of the information-theoretic perspective. Below, we address your remaining concerns.
>
> **Despite its strong conceptual foundation, the paper’s methodological presentation is somewhat difficult to follow. Understanding the full framework requires frequent reference to the appendix, which makes it challenging to reconstruct the complete algorithmic pipeline from the main text. In particular, Step 2 and Step 10 of Algorithm 1, which concern the estimation of the missingness mechanism and its integration into the weighting scheme, are crucial for comprehension and should be explicitly detailed in the main body.**
>
> Thank you for raising this lack of clarity. We have reworked the manuscript and used the additional page in the final version to provide further details on the proposed methodology. Please see the updated section 4, which now provides an overview of the algorithm.
>
>
> **It also remains unclear how exactly this missingness model is parameterized or estimated in practice.**
>
> **Could you elaborate on how $p_{}$ is actually modeled and estimated? How sensitive is performance to misspecification of this model?**
>
> Thank you for raising this concern. In our experiments with clinical data and MNAR-simulated datasets, we propose parametrizing the propensity model using logistic regression. We have clarified this point in the manuscript by adding Appendix G.2 to describe the propensity model in the real-world analysis. As you correctly noted, misspecification of this model can lead to bias in the propensity weights and, consequently, performance estimates. Critically, one must control the covariates to satisfy the MAR. Beyond the scope of our study, [1] studies the impact of misspecification on the IPW estimate. Practically, we can use any flexible model class as long as the models are calibrated, which we ensure in our experiments. Heuristically, it is recommended to threshold the propensities within a user-defined threshold to ensure positive support.
>
> [1] Austin, Peter C., and Elizabeth A. Stuart. "The performance of inverse probability of treatment weighting and full matching on the propensity score in the presence of model misspecification when estimating the effect of treatment on survival outcomes." *Statistical methods in medical research* 26.4 (2017): 1654-1670.
>
> **Moreover, the objective of minimizing appears counterintuitive and should be clarified, as mutual information is typically maximized in multimodal fusion.**
>
> Thank you for raising this point. As you noted, one often aims to maximize mutual information when training predictive models. Our goal is not predictive modelling but Partial Information Decomposition (PID). Appendix B shows how obtaining $q$ that *minimizes* the three-way mutual information corresponds to estimating the PID: shared, unique 1, unique 2, and complementary information, as demonstrated by Bertschinger et al. [1]. More specifically, we are minimizing the three-way mutual information under distribution $q(y, x_1, x_2)$, a distribution that matches the two-way *actual* joints $p_{\Omega}(y, x_1)$ and $p_{\Omega}(y, x_2)$, but minimizes  $I_q(Y:(X_1,X_2))$ to recover $CI(Y:X_1;X_2)=I_{\Omega}(Y:(X_1,X_2))$ $-min_{q \in \Delta_{\Omega}}I_q(Y:(X_1,X_2))$, and consequently all other bounds. For a more detailed explanation, please refer to Lemma 4 in “Quantifying Unique Information” [1]. To address your concern, we have updated Section 4 to clarify the proposed algorithm and updated Appendices B and C.
>
> [1] Bertschinger et al. "Quantifying unique information." *Entropy* (2014).

---

> ### Author Response · Authors · 2025-11-21
>
> **From an empirical perspective, while the synthetic and semi-synthetic experiments are carefully designed and demonstrate the motivation, the work lacks convincing real-world evidence that the proposed correction improves downstream predictive performance. Correcting modality informativeness estimates is valuable, but readers are left wondering whether a practitioner facing genuinely missing modalities would actually see performance gains by applying ICYM2I.**
>
> Thank you for raising this concern. While our work proposes a new strategy to identify which covariates lead to the best performance under fully observed covariates at deployment, it does not aim to improve predictive performance, but rather to provide an unbiased estimate of information gain of using a modality, and an unbiased estimate of a multimodal model’s performance that uses the modality at deployment. The utility of our method is that practitioners can garner proper estimates of both model performance and modality informativeness. Our work should be seen as a toolbox that can help practitioners determine whether a modality is worth collecting, including whether joint modeling is useful. This may not seem valuable for a web-scale vision-language model, but it is crucial when collecting modalities incurs costs, such as in decisions to include additional sensors for autonomous vehicles or to recommend that clinicians run another expensive diagnostic exam. We have clarified this point throughout the manuscript. Indeed, our clinical example clearly demonstrated that collecting chest radiographs is unwarranted for detecting structural heart disease compared to ECGs, which is the intended use of the proposed method.
>
> **Finally, the framework’s dependence on a particular parameterization $q_{\theta}=\exp{}(f_1,f_2)$ is restrictive; demonstrating that the approach generalizes across different fusion architectures would strengthen the empirical case and enhance credibility.**
>
> Thank you for raising this point. First, fusion in the multimodal sense refers to any model that uses multiple modalities, which we have referred to as $p(y| x_1, x_2)$ in our manuscript. The parametrization you mentioned is intended only for $\text{ICYM}^2\text{I-PID}$ to recover the partial information decomposition (PID) quantities [1]. The method requires an approximation of a three-dimensional joint probability tensor $q$ where the two-way joint probabilities between each input modality $X_i$ and label $Y$ match the true distribution $p$, i.e., $q(x_i,y)=p_{actual}(x_i,y)$ as required by the bounds from Bertschinger et al. [1]. The focus of our method is on the necessary corrections to obtain an unbiased PID estimate. Our framework is agnostic to the parametrization of $q$. While there may exist parametrizations that offer better inductive biases for specific tasks, our framework itself is a flexible plug-and-play for any parametrization that provides reliable logits. We focus our contributions on the correction and, therefore, rely on the choice from prior works that do not focus on the need for correction [2,3], which provides a scalable and flexible method of approximating $q$ from high-dimensional inputs.
>
> We also note that the learning component of our work, $\text{ICYM}^2\text{I-learn}$, is architecture-agnostic, and practitioners can use any fusion model that produces calibrated probabilities. Other parameterizations for $q$ are an avenue for future work since we think these should primarily be application-driven. Our parametrization, drawn from Liang et al, is certainly flexible enough for all real-world datasets tested in this work. We have clarified this point in Appendix C of our revised manuscript.
>
> [1] Bertschinger et al. "Quantifying unique information." *Entropy* (2014).
> [2] Liang, Paul Pu et al. "Quantifying & modeling multimodal interactions: An information decomposition framework." *Advances in Neural Information Processing Systems* 36 (2023): 27351-27393.
> [3] Tsai, Yao-Hung Hubert et al. "Multimodal transformer for unaligned multimodal language sequences." In *Proceedings of the conference. Association for computational linguistics. Meeting*, vol. 2019, p. 6558. 2019.
>
> **Lastly, a weakness regarding the submission is that the code has not been shared, either in the supplementary material or via an anonymized GitHub repository, which limits reproducibility and verification of the claims.**
>
> Thank you for bringing this critical concern about reproducibility. The manuscript contains an anonymized link to the GitHub repository on page 1 (https://anonymous.4open.science/r/ICYM2I-BC18/). As this footnote may have been easily missed, we clarified the availability of the code in the text.

---

> ### Author Response · Authors · 2025-11-21
>
> **Have you considered cases with unpaired modalities, where instances across modalities are not aligned? Could you comment how the method could be extended to such cases?**
>
> Thank you for this suggestion. We believe ‘unaligned’ or ‘unpaired modalities’, as you suggest, refers to the notion that available modality is not tied to a specific data sample or instance. Extending this method when there is no notion of an instance is plausible for both prediction and PID but requires different inductive biases to model the underlying unimodal and multimodal probabilities, i.e., that can learn from unaligned samples as context. The robustness of such models needs further exploration and possible innovation. In the current method, we assess informativeness when modality is missing at random, assuming that each instance $i$, e.g. a patient, is associated with the tuple $(X_{1,i}, X_{2,i}, Y_i)$ where one modality may be missing. For example, our real-world study, a chest X-ray (CXR) and an electrocardiogram (ECG) can be observed for each patient when detecting structural heart disease, but a modality may be missing at random. We hope this clarifies your query. If we misunderstood what you mean by 'unaligned,' please let us know, and we are happy to clarify further.
>
> **Would the method still hold for other forms of fusion beyond, e.g. attention-based models especially when large in parameter number?**
>
> Thank you for raising this point. We note that our framework is agnostic to the choice of fusion architecture and is certainly applicable beyond attention-based models. As long as the user can acquire calibrated prediction probabilities, our method will return adjusted predictive performance and information gain estimates. We will further clarify this in the introduction of our updated manuscript.
>
> **Could one imagine a generative formulation where C and Y are governed by a single latent variable, and would this affect how modality informativeness is estimated?**
>
> Thank you for this question. If C and Y have a latent or unobserved common parent, the missingness mechanism is MNAR and would indeed violate the MAR assumption, leading to a biased estimate of informativeness without making strong distributional and untestable assumptions on the latent variable. Our paper’s core contribution is to formalize this problem of missingness in *multimodal settings* and introduce a correction in the realistic MAR setting. Unfortunately, broadly, the problem of MNAR missingness is highly challenging to address without strong assumptions.
>
> We hope these proposed changes address your concerns and that you will consider increasing your score.

---

> > ### Comment · Reviewer_VFgX · 2025-11-22
> >
> > Thank you for taking the time to answer the questions. I would be glad to see the changes. I would propose as well the question regarding unpaired modalities to be included somewhere in the discussion and a short ablation study with a couple of more fusion methods to be included in the appendix on the camera ready version. I will maintain my score and confidence.

---

> > > ### Author Response · Authors · 2025-11-23
> > >
> > > Thank you for the additional suggestions. We have expanded Section 4.3 to discuss the agnosticity to choice of parametrisation, and we now address the issue of unpaired modalities in the Discussion. All changes are included in red in the uploaded revised version.

---

### Official Review · Reviewer_7Hat · 2025-10-31

**Soundness:** 3
**Presentation:** 2
**Contribution:** 3
**Rating:** 6
**Confidence:** 2

**Summary:**

The paper studies an issue in multimodal learning: when training/evaluating only on samples where all modalities are present, we bias both performance estimates and conclusions about each modality’s "informativeness".

The authors formalize missingness as a distribution shift and propose $\text{ICYM}^2\text{I}$, a double inverse-probability weighting (IPW) correction—apply IPW at training time and again at evaluation time—to recover estimates with respect to the target (fully observed) population.

They further adapt Partial Information Decomposition (PID) to high-dimensional settings to decompose a modality’s contribution into unique, shared, and complementary/synergistic information, and show that naive PID computed on complete-case data can be substantially biased.

**Strengths:**

- **Methodologically sound correction**. Leveraging IPW for both training and evaluation addresses two common sources of bias.
- **Information-theoretic perspective**. Connecting PID to missingness is insightful; prior PID work [1] did not consider selection bias from partial observation.
- **Thorough experiments**. The synthetic/XOR table shows great insights about how complete-case analysis can invert conclusions (even negative "shared" due to bias) and how correction aligns with oracle; this pattern recurs in UR-FUNNY and Hateful Memes.

[1] Williams, Paul L., and Randall D. Beer. "Nonnegative decomposition of multivariate information." arXiv preprint arXiv:1004.2515 (2010).

**Weaknesses:**

- **Assumption strength and scope**. The core guarantee hinges on MAR + positivity. I wonder whether this holds true in most multimodal systems. For instance, missingness can be MNAR, which violates the MAR assumption. The paper simulates MNAR in the appendix but does not probe MNAR in the clinical study. I suggest authors add sensitivity analyses on real-world data.
- **Breadth of baselines**. Empirically, the paper compares observed vs. corrected vs. oracle. It would help to include strong missing-modality baselines (e.g., SMIL [2], modality-incomplete prompt/adapters [3]) to situate ICYM2I among robustness methods; also relate to missingness-shift DA.

[2] Ma, Mengmeng, et al. "Smil: Multimodal learning with severely missing modality." Proceedings of the AAAI conference on artificial intelligence. Vol. 35. No. 3. 2021.

[3] Lee, Yi-Lun, et al. "Multimodal prompting with missing modalities for visual recognition." Proceedings of the IEEE/CVF Conference on Computer Vision and Pattern Recognition. 2023.

**Questions:**

Please refer to Weaknesses

---

> ### Author Response · Authors · 2025-11-21
>
> Thank you for your thoughtful and constructive feedback. We appreciate your positive assessment of the methodological soundness of our approach, the relevance of extending Partial Information Decomposition (PID) to settings with partial observation, and the value of our experimental analysis. Below, we address the concerns raised and provide further clarification.
>
> ***Assumption strength and scope.* The core guarantee hinges on MAR + positivity. I wonder whether this holds true in most multimodal systems. For instance, missingness can be MNAR, which violates the MAR assumption. The paper simulates MNAR in the appendix but does not probe MNAR in the clinical study. I suggest authors add sensitivity analyses on real-world data.**
>
> Thank you for highlighting the importance of the missing at random (MAR) and positivity assumptions. As you noted, violations of these assumptions may lead to biased performance and information-gain estimates. A core contribution of our work is the formalization of the problem of missingness in multimodal settings. Through this formalization, we highlight the assumptions under which one can address the biases associated with missingness. We believe that transparency in outlining these assumptions is critical to improving the applicability of these strategies and a limitation of methods that often overlook the complexity and type of multimodal missingness.
>
> Further, as you noted, Appendix D includes a sensitivity analysis with simulations for both the missing completely at random (MCAR) and missing not at random (MNAR) settings. In the MCAR setting, all methodologies yield comparable performance estimates, demonstrating the stability of our method under both MAR and MCAR settings. However, when the missingness of a variable is drawn as a function of the variable itself (MNAR), we observed biased performance and information estimates. Under such missingness patterns, the correction is not guaranteed to eliminate the bias in estimates because the missingness process cannot be modelled from observed covariates, and therefore cannot be corrected.
>
> Regarding performing a sensitivity analysis on our clinical dataset, we note that the structural heart disease (SHD) data already contains inherent missingness; not everyone who gets an ECG gets an X-ray based on various factors. Therefore, it is not possible to explore changes in estimates under alternative missingness patterns, as one cannot enforce other realistic MNAR missingness patterns. For this reason, controlled sensitivity analysis can only be synthetically induced on real data (such as the one presented in Appendix D) and is the standard method for assessing robustness to MAR violations (e.g. [1,2]). To address your concern in semi-synthetic settings, we have included an additional experiment (Appendix F.1.2) on the UR-FUNNY real-world dataset, in which missingness is generated based on the missing modality itself. This MNAR setting echoes the results of the logic experiment, in which failing to meet the MAR assumption leads to biased estimates.
>
> To further address your concern, we have clarified the limitations of MAR/MNAR identifiability in the conclusion.
>
> *“The key assumption in our work is that missingness is MAR. No theoretical guarantees exist under MNAR patterns. While distinguishing these assumptions is empirically untestable, practitioners should ensure that this assumption is appropriate for their data. Importantly, MAR is less restrictive than the implicit MCAR assumption made in the multimodal literature, and does not require unrealistic distributional assumptions that one must assume to tackle MNAR patterns.”*
>
> [1] Nabi, Razieh, and Rohit Bhattacharya. "On testability and goodness of fit tests in missing data models." *Uncertainty in Artificial Intelligence*. PMLR, 2023.
> [2] Ji, Feng, Sophia Rabe-Hesketh, and Anders Skrondal. "Diagnosing and handling common violations of missing at random." *psychometrika* 88.4 (2023): 1123-1143.

---

> ### Author Response · Authors · 2025-11-21
>
> ***Breadth of baselines.* Empirically, the paper compares observed vs. corrected vs. oracle. It would help to include strong missing-modality baselines (e.g., SMIL [2], modality-incomplete prompt/adapters [3]) to situate ICYM2I among robustness methods; also relate to missingness-shift DA.**
>
> Thank you for this suggestion. We agree that SMIL [1] and modality-incomplete prompting and adapters [2] are valuable approaches that improve model robustness when a modality is missing completely at random. Specifically, Ma et al. [1] and Lee et al. [2] randomly drop modalities at a fixed percentage, inducing MCAR missingness patterns in training, implying that the models will only work if the test data is also MCAR. Our work studies a more realistic missingness process, as observed in domains like healthcare, where missingness is driven by decision rules, costs, and potential causal relations between modalities.
>
> In this context, the goal of our method ICYM^2I is not to introduce a new method for robustness to missingness; rather, it is to provide unbiased estimates of *predictive performance and information gain under MAR*, correcting for the distribution shift between modality-complete samples and the overall sample distribution. In this sense, our work is related to missingness-shift DA [3]. Our work is the first to formalize multimodal missingness, connecting it to distribution shift at deployment, and provides a corrected estimate of performance under such a shift. Methods such as SMIL [1], and Lee et al [2] can be complementarily assessed using our proposed corrected evaluation.
>
> If the test data are MCAR, both SMIL[1] and Lee et al. [2] would provide an unbiased performance estimate on MCAR test data, and so would our predictive method. We compare these performance estimates using the Hateful Memes dataset with the multimodal prompting method introduced by Lee et al. [2] under MAR missingness patterns. We perform a lightweight evaluation by training a vision-language transformer (ViLT) on the full training set. Then, we enforce between 30% to 70% of missingness on the text modality based on the image modality, and evaluate AUROC on the test set. We train the model for 10 epochs.
>
> Our preliminary results demonstrate that predictive performance estimates are not improved by this robustness intervention under MAR patterns, as existing robustness methods are not developed for such patterns. In contrast, our proposed approach yields performance closer to the oracle's, demonstrating improved performance when shifting from training with missingness to a fully observed test set.
>
> Finally, even under MCAR, existing robustness methods rely on extracting high-dimensional representations through imputation or reconstruction of missing modalities or prompt embeddings, leading to unstable estimates. In contrast, our method requires only estimating unidimensional propensity weights, a simpler task that results in more stable estimates.
>
> [1] Ma, Mengmeng, et al. "Smil: Multimodal learning with severely missing modality." *Proceedings of the AAAI conference on artificial intelligence*. Vol. 35. No. 3. 2021.
> [2] Lee, Y. L., et al. (2023). Multimodal prompting with missing modalities for visual recognition. In *Proceedings of the IEEE/CVF Conference on Computer Vision and Pattern Recognition* (pp. 14943-14952).
> [3] Zhou, Helen, et al. "Domain adaptation under missingness shift." *International Conference on Artificial Intelligence and Statistics*. PMLR, 2023.
>
> ---
> We sincerely appreciate the reviewer’s insightful feedback on our work. We hope that the clarifications provided address your concerns and further highlight the significance of our contribution. If the reviewer would consider revisiting the evaluation in light of these revisions, we would be deeply grateful.

---

> > ### Author Response · Authors · 2025-11-23
> >
> > ***Breadth of baselines.* Empirically, the paper compares observed vs. corrected vs. oracle. It would help to include strong missing-modality baselines (e.g., SMIL [2], modality-incomplete prompt/adapters [3]) to situate ICYM2I among robustness methods; also relate to missingness-shift DA.**
> >
> > We thank the reviewer again for their suggestion. As noted in our previous response, both SMIL [1] and modality prompting [2] address and experiment with missing completely at random (MCAR) missingness, while our setting is for missing at random (MAR) settings where missingness can be modelled using observed covariates. To help illustrate this distinction, we performed a comparison using the multimodal prompting method from Lee et al. [2] under MAR–induced text missingness rates of 30 to 70%:
> >
> > | Ratio (%) | ViLT Prompt            | ICYM²I                   | ViLT Prompt Δ | ICYM²I Δ |
> > |-----------|-------------------------|---------------------------|----------------|-----------|
> > | Oracle    |                         | 0.575 (0.539–0.611)       |                |           |
> > | 30        | 0.570 (0.534–0.606)     | 0.567 (0.530–0.601)       | -0.005         | -0.008    |
> > | 40        | 0.545 (0.511–0.578)     | 0.567 (0.533–0.604)       | -0.030         | -0.008    |
> > | 50        | 0.560 (0.522–0.593)     | 0.567 (0.533–0.602)       | -0.015         | -0.008    |
> > | 60        | 0.560 (0.526–0.593)     | 0.562 (0.526–0.597)       | -0.015         | -0.013    |
> > | 70        | 0.535 (0.497–0.571)     | 0.573 (0.536–0.609)       | -0.040         | -0.002    |
> >
> > Across different missingness levels, our method, $\text{ICYM}^2\text{I}$, remains generally closer to the oracle AUROC value than the prompting method, despite the performance of the ViLT-based method at higher levels of missingness.
> >
> > We hope this additional comparison clarifies how our contribution is a complementary problem to robustness to modality missingness in recovering unbiased performance estimates under missingness.
> >
> > [1] Ma, Mengmeng, et al. "Smil: Multimodal learning with severely missing modality." *Proceedings of the AAAI conference on artificial intelligence*. Vol. 35. No. 3. 2021.
> > [2] Lee, Y. L., et al. (2023). Multimodal prompting with missing modalities for visual recognition. In *Proceedings of the IEEE/CVF Conference on Computer Vision and Pattern Recognition* (pp. 14943-14952).

---

### Author Response · Authors · 2025-12-04

We thank the reviewers for their insightful feedback and the area chairs for their facilitation. Our paper makes the following contributions:

1. We highlight the **importance of accounting for missingness in multimodal learning** and formalize the problem **through the lens of distribution shifts**, previously ignored in multimodal literature.
2. We propose $\text{ICYM}^2\text{I}$, a **principled framework for accounting for multimodal missingness** grounded in statistical missingness.
3. We introduce an **inverse-probability weighting (IPW) framework** for unbiased training and evaluation of multimodal models under missing at random (MAR) settings in $\text{ICYM}^2\text{I}$-learn.
4. We develop an **autodifferentiable method for evaluating information gain** with partial information decomposition (PID) under MAR with $\text{ICYM}^2\text{I}$-PID.
5.  We conduct **empirical evaluation across diverse multimodal settings**, including affective computing, online content understanding, and healthcare. Additionally, we perform ablations and sensitivity analyses to assess robustness under MAR violations in both missing completely at random (MCAR) and missing not at random (MNAR) settings.

The reviewers found strong agreement that the identified problem of multimodal missingness is an important and underexplored problem, that our formulation is meaningful, and that the proposed framework is methodologically sound. We have addressed the reviewers’ concerns and strengthened our contributions as summarized below:

- *MAR assumption and sensitivity analysis.* To address reviewer 7Hat and pctf’s concerns around violations to the MAR assumption, we expanded the discussion around the applicability and implications of this assumption, as well as the challenges around methods that test for MAR violations in the conclusion of our paper. We highlight that addressing MNAR remains a challenging and open question, and that knowledge of the missingness mechanism, typically never available, is needed to overcome MNAR multimodal missingness. We added additional sensitivity analyses, including on semi-synthetic MNAR in Appendix F.1.2, and clarified how violations of MAR affect both the predictive performance and information-gain estimates.

- *Missingness mechanism estimation.* In response to reviewer VFgX and pctf’s request for more clarity on how the missingness mechanism is modeled, we added an expanded description of how the propensity model is parameterized and estimated in practice, including the logistic regression propensity models used in our experiments with real-world data in Appendix G.2. We further clarified the key steps in our algorithm in the final page of the manuscript.

- *Scope, generalizability, and implications of our framework.*  We demonstrate that multimodal missingness arises across real-world domains and is often overlooked in dataset construction and evaluation despite its impact. By grounding the problem in statistical missingness and distribution shifts, we clarify the limits of existing robustness approaches and the necessity of correcting for MAR-induced bias. Reviewer pctf raised questions on multimodal settings beyond two input modalities. We note that $\text{ICYM}^2\text{I}$-learn naturally extends to an arbitrary number of input modalities under MAR, and that $\text{ICYM}^2\text{I}$-PID can be applied in a one-versus-rest manner when practitioners wish to assess the informativeness of individual modalities, as extending PID beyond pairwise interactions remains an open challenge in information theory. We expanded the future work section of our manuscript in response to potential extensions.

We thank the reviewers and area chairs for their consideration.

---

### Meta-Review · Area_Chair_L3BA · 2025-12-23

**Summary:**

The paper reveals an issue of underlying domain shift between observed and missing data, which results in biases in modality informativeness estimates. It further introduces ICYM2I framework to correct this bias by employing Inverse Probability Weighting to adjust observed data weights. Experiments on synthetic and real-world data indicate that ICYM2I can alleviate the bias of modality informativeness estimates caused by missingness-induced domain shift. However, the manuscript can be enhanced by a clearer emphasis that ICYM2I does not lead to improved predictive performance, but serves as a toolbox to convey more accurate estimation of modality importance and informativeness, which limits the scope and real-world applicability of the proposed framework. Additionally, a more complete comparison (with information decomposition result) with more baselines suggested by the reviewer may further improve the submission. Given that most reviewers hold positive scores for the submission, my recommendation would be weak acceptance.

**Reviewer Concerns:**

The concerns raised by reviewers are as follows.

Reviewer 7Hat  raises two concerns: assumption strength (the main assumption of ICYM2I is MAR, where missingness is dependent on the observed variable) and missing baseline. For the former concern, the authors claimed that the scope of the paper remains in the MAR assumption since MNAR is a more challenging scenario. As for the latter, the authors conducted further experiments on additional datasets; in the additional experiments, the information decomposition result is not reported, while this can be critical since the main contribution and aim of ICYM2I is to debias the informativeness estimate, instead of improving predictive performance.

Reviewer VFgX's concerns include presentation (crucial elaboration are lacked in the main body, e.g., how to estimate p in the correction term); why minimize mutual information instead of maximizing; lack of evidence for correction improves predictive performance, real-world application of the proposed framework remains unclear, and reproductibility issue without open-sourced codes. During the rebuttal, some concerns are addressed, and the manuscript is revised accordingly. On the other hand, the authors agree that ICYM2I does not aim to improve performance and has no guarantee of superior performance, indicating limited real-world application to some extent.

Rviewer Pctf also raised concerns about the assumption strength, modality scalability, and the limited applicability of experiments and generalization. The authors clarified that the clinical data are from multiple institutes and expanded the discussion on cases involving more than two modalities.

**Reviewer Scores:**

I think the reviewers would tend to keep their scores if they fully participate in the discussion.

---

### Decision · Program_Chairs · 2026-01-26

Accept (Poster)